



**Dynamic MJO forecasts using an ensemble subseasonal-to-seasonal**
**forecast system of IAP-CAS model**
Yangke Liu[1,6], Qing Bao[*1], Bian He[1], Xiaofei Wu[2], Jing Yang[3], Yimin Liu[1], Guoxiong Wu[1], Tao Zhu[1],
Siyuan Zhou[1], Yao Tang[1,6], Ankang Qu[1,7], Yalan Fan[3], Anling Liu[3], Dandan Chen[1,6], Zhaoming Luo[1,7],
Xing Hu[4], Tongwen Wu[5]
[1]State Key Laboratory of Numerical Modeling for Atmospheric Sciences and Geophysical Fluid Dynamics (LASG), Institute
of Atmospheric Physics, Chinese Academy of Sciences, Beijing 100029, China
[2]School of Atmospheric Sciences/Plateau Atmosphere and Environment Key Laboratory of Sichuan Province, Chengdu
University of Information Technology, Chengdu 610225, China
[3]Faculty of Geographical Science, Beijing Normal University, Beijing 100875, China
[4]National Meteorological Information Center, China Meteorological Administration, Beijing 100081, China
[5]Center for Earth System Modeling and Prediction, China Meteorological Administration, Beijing 100081, China
[6]College of Earth and Planetary Sciences, University of Chinese Academy of Science, Beijing 100049, China
[7]School of Emergency Management Science and Engineering, University of Chinese Academy of Science, Beijing 100049,
China
*Correspondence to*: Qing Bao (baoqing@mail.iap.ac.cn)



**Abstract.** The Madden-Julian Oscillation (MJO) is a crucial predictability source on a sub-seasonal to seasonal (S2S) timescale.
Therefore, the models participating in the WWRP/WCRP S2S prediction project focus on accurately predicting and analyzing
the MJO. This study provided a detailed description of the configuration within the IAP-CAS S2S forecast system. We assessed
the accuracy of the IAP-CAS model's MJO forecast using traditional RMM analysis and cluster analysis. Then, we explained
the reasons behind any bias observed in the MJO forecast. Comparing the 20-year hindcast with observations, we found that
the IAP-CAS ensemble mean has a skill of 24 days. However, the ensemble spread still has potential for improvement. To
examine the MJO structure in detail, we used cluster analysis to classify the MJO events during boreal winter into four types:
fast-propagating, slow-propagating, standing, and jumping patterns of MJO. The model exhibits biases of overestimated
amplitude and faster propagation speed in the propagating MJO events. Upon further analysis, it was found that the model
forecasted a wetter background state. This leads to more intense forecasted convection and stronger coupled winds, especially
in the fast MJO events. However, the horizontal moisture advection effect for eastward propagation is overestimated in IAP-
CAS due to the wetter state and more substantial MJO circulations, which results in a faster MJO mode. These findings show
that the IAP-CAS skilfully forecasts signals of MJO and its propagation, and they also provide valuable guidance for improving
the current MJO forecast by developing the ensemble system and moisture forecast.
**Keywords**: MJO prediction, S2S, IAP-CAS, FGOALS-f2, Cluster Analysis



**1 Introduction**

With the increasing occurrence of metrological disasters in recent years, there has been growing attention toward S2S forecast,
as it bridges the gap between weather and climate forecasts and reduces disaster risks through early warnings. In November
2013, the WWRP/WCRP S2S prediction project (Phase 1) was launched,   with the principal objectives of enhancing S2S
forecast accuracy and advancing our comprehension of its dynamics and climate drivers.   Then, work on the S2S research
continued in Phase 2, from 2018 to 2023. The whole project has made a significant contribution to the development of S2S
prediction.
MJO (Madden and Julian, 1971) is a crucial predictability source of S2S forecasts. It is a significant tropical oscillation with
a period of 30-60 days, characterized by expansive cloud masses and precipitation systems that propagate eastward along the
equatorial regions. Accurate S2S prediction requires a good representation of MJO. Many studies have clarified the relationship
between the MJO and global weather and climate, such as monsoons (Goswami, 2012; Hsu, 2012; Lau and Chan, 1986;
Wheeler et al., 2009), tropical cyclones (Bessafi and Wheeler, 2006; Ferreira et al., 1996; Hall et al., 2001) and El Niño-
Southern Oscillation (ENSO; Lau et al., 2005; Zhang, 2005). The convective and circulation anomalies associated with MJO
establish intricate connections across global weather and climate systems on the S2S timescale. Being able to accurately
forecast the MJO can have a positive impact on the forecast of other related systems (Cassou, 2008; Vitart and Molteni, 2010;
Wu et al., 2007). Achieving an accurate forecast of MJO becomes a primary objective in the field of S2S forecasts.
With an enhanced comprehension of the underlying physical mechanisms governing the MJO and the continuous improvement
of numerical models, remarkable advancements have been achieved in the MJO forecast. In Coupled Model Intercomparison
Project Phase 6 (CMIP6), models that exhibited lower forecast skills (Hung et al., 2013) in Coupled Model Intercomparison
Project Phase 5 (CMIP5) have demonstrated noteworthy improvements in the simulation of MJO (Ahn et al., 2020). Generally,
the models in CMIP6 simulate more realistic eastward propagation and precipitation over the Maritime Continent (MC) region
(Ahn et al., 2019; Chen et al., 2022).
However, for S2S forecasts, the improvement of model physics is one aspect of advancing S2S forecasts, as various factors
impact MJO forecast skills, such as initialization and ensemble generation (Kim et al., 2018). The forecast skills of the MJO
in most models is typically 3-4 weeks (Vitart, 2017), while the estimate of predictability of MJO is approximately 5-7 weeks
(Waliser et al., 2003; Neena et al., 2014). These facts underscore the persisting challenges in the S2S forecasts.
The realistic forecast of MJO eastward propagation is one of the challenges repeatedly mentioned in recent years (Jiang, 2017;
Kim, 2019; Lim et al., 2018; Wang and Lee, 2017). The MJO propagation skill is closely related to the forecast of the state in
the Maritime Continent (MC) region (Gonzalez and Jiang, 2017). Many studies have pointed out the "MC barrier" (Hendon
and Salby, 1994; Rui and Wang, 1990a; Vitart et al., 2017) during the MJO's propagation through the MC region. The "MC
barrier" refers to a notable deterioration of the MJO signal when it traverses the MC area, but this phenomenon is usually



amplified in the climate models (Kim et al., 2014b; Neena et al., 2014; Xiang et al., 2022, 2015), showing the model's limitation
in preserving MJO propagation within the MC region. The moisture mode theory (Raymond and Fuchs, 2009) has been
proposed to explain this phenomenon. It suggests that the advection of seasonal mean moisture by the MJO-related circulation
anomalies in the low troposphere is crucial to MJO's propagation through the MC region (Jiang, 2017; Kim, 2019). In models
that are hard to capture the realistic propagation of MJO, the mean low-troposphere moisture amplitude over the MC is
underestimated, resulting in a weakened horizontal moisture gradient (Gonzalez and Jiang, 2017; Kim, 2017). This discrepancy
in moisture advection hinders MJO propagation.
The Institute of Atmospheric Physics at the Chinese Academy of Sciences (IAP-CAS) has been actively involved in climate
model development and applications since the CMIP1 in the 1990s. As for the IAP-CAS model, it has already shown a
significant enhancement in MJO simulation in CMIP6 compared to CMIP5 (Chen et al., 2022), but the performance of the
S2S system in IAP-CAS remains uncertain and requires comprehensive evaluation. Therefore, the objectives of this article are
fourfold: Firstly, the aim is to introduce the S2S forecast system of the IAP-CAS model. Secondly, to evaluate the forecast
skills of the IAP-CAS in the MJO forecast. Thirdly, the aim is to analyze the evaluation results to identify the sources of
forecast errors. This will facilitate further improvements in the MJO forecast. At last, we hope that the verification and analysis
process can provide some valuable insights for other models.
The structure of the paper is as follows. A thorough review of the IAP-CAS model and S2S ensemble forecast system is
introduced in Section 2. Section 3 describes the observation data and primary methodology utilized in the article. Section 4
assessed the overall MJO forecast skills in IAP-CAS. Section 5 focused on analyzing the propagation details of the fast-
propagating and slow-propagating MJO. After that, in Section 6, we discussed the potential causes of any bias observed in the
MJO forecast. In Section 7, we summarized our findings and had a discussion.
**2 The global S2S ensemble forecast system of IAP-CAS**
The architecture of the IAP-CAS S2S ensemble forecast system is depicted in Figure 1. In this section, we will give a thorough
description of the S2S system, covering the model, initialization methods, ensemble generation approaches, and the resulting
datasets.
**2.1 Configuration of IAP-CAS model**
The climate system model CAS FGOALS-f2 (The Flexible Global Ocean-Atmosphere-Land System model Finite Volume
version 2, Chinese Academy of Sciences; Bao 2019; Bao et al. 2020) is the core of the IAP-CAS S2S ensemble forecast system.
It is developed by the State Key Laboratory of Numerical Modeling for Atmospheric Sciences and Geophysical Fluid
Dynamics (LASG) at the Institute of Atmospheric Physics (IAP), Chinese Academy of Sciences (CAS). We utilize the
institution name, IAP-CAS, as a proxy for the model.



FGOALS-f2 is a fully coupled model that encompasses four components: atmospheric, land, oceanic, and sea ice models, with
its configuration detailed in Table 1. The atmospheric component is version 2 of the Finite-volume Atmospheric Model
(FAMIL2; Li et al. 2019), with a standard horizontal resolution of C96, which means 96 × 96 grid points in each tile of the
cube sphere, roughly equivalent to 1-degree resolution. Vertically, it features 32 hybrid sigma-pressure levels, with the
uppermost level situated at 1 hPa (The Hybrid coefficients are listed in Table 2). The land surface component used in FGOALS-
f2 is version 4 of the Community Land Model (CLM4.0; Oleson et al. 2010; Lawrence et al. 2011), featuring a horizontal
resolution nearly at 1-degree resolution. The oceanic component is Parallel Ocean Program version 2 (POP2; Kerbyson and
Jones 2005), which utilizes a displaced-pole grid with the North Pole shifted to Greenland. This grid has a resolution of gx1v6,
approximately equivalent to a 1-degree horizontal resolution, and includes 60 vertical layers. The sea ice component is the Los
Alamos Sea Ice Model version 4.0 (CICE4; Hunke et al. 2010), sharing the exact horizontal resolution as the ocean model.
These four components are coupled via the coupler version 7 in the Community Earth System Model (CESM; Craig et al.

104    2012).

It is worth noting that FAMIL2, the latest generation atmospheric model from LASG, has adopted the Finite-Volume Cubed-
Sphere Dynamical Core (FV3; Lin 2004; Putman and Lin 2007) as its dynamical core. FV3 solves the fully compressible Euler
equations on the gnomonic cubed-sphere grid and a Lagrangian vertical coordinate. Fast vertically propagating sound and
gravity waves are solved by the semi-implicit method (Harris et al., 2020). This enhancement of the atmospheric component
results in improved computational efficiency and accuracy. Besides, the key parameterization in FAMIL2 is a Resolved
Convection Precipitation scheme (RCP), which is independently developed to calculate the microphysics processes in the
convective precipitation for both deep and shallow convection (Bao and Li, 2020). Due to the rapid phase changes occurring
within the convective cloud, a sub-time step of 150 seconds is employed for the calculation of microphysical processes within
a physical timestep of 30 minutes. FAMIL2 has also implemented the University of Washington Moist Turbulence
parameterization scheme (UWMT, Park and Bretherton 2009) as its boundary layer scheme. The microphysical
parameterization used in FAMIL2 is the revised Lin scheme, which is a single-moment scheme (Zhou et al., 2019).
Building upon previous work, it has been observed that the IAP-CAS model can effectively reproduce the patterns and intensity
of ENSO variability (Fig. A2), and it also shows a skillful forecast of tropical cyclones (Li et al., 2021, 2022). As mentioned
in Section 1, the activity of the MJO is significantly impacted by the tropical weather and climate systems, including ENSO
and tropical cyclones (TCs). The accurate forecast of ENSO and TCs heightens our anticipation for MJO forecast skill in the
IAP-CAS model. FAMIL2 has already provided a realistic forecast of convectively coupled equatorial waves (CCEWs) and
MJO convection (Li et al., 2019). The S2S forecast system also exhibits excellent performance in forecasting arctic sea ice
(Liu et al., 2023) and spring rainfall (Fan et al., 2023).





## 2.2 Initialization scheme of the S2S forecast system

The S2S forecast system of the IAP-CAS model adopts a Newtonian nudging method with time-varying treatment (Jeuken et al., 1996) to complete the initialization of the atmosphere and ocean. The reanalysis nudging and the forecast nudging are the two components that make up the initialization process, which is seen in Figure 2. Table 3 provides a summary of the detailed technical specifics for these two nudging processes.

The reanalysis nudging initializes the atmospheric variables, including temperature, surface pressure, sea level pressure, and surface wind from the NCEP Final Operational Global Analysis datasets (FNL, http://rda.ucar.edu/datasets/ds083.2, ds083.2|DOI: 10.5065/D6M043C6). The oceanic variable of potential temperature from the National Oceanic and Atmospheric Administration (NOAA) Optimum Interpolation Sea Surface Temperature (OISST) reanalysis data (Reynolds et al., 2007) is also included. These reanalysis data serve as observations in the eq. (1) to diminish errors in the initial condition:

$$x(t) = x_{model}(t) + N_{rea}(t)[x_{obs}(t) - x_{model}(t)] \tag{1}$$

where $t$ is the time, $x(t)$ is the filed after nudging process, $x_{model}(t)$ represents the model forcing, $x_{obs}(t)$ represents the "truth" value, and $N_{rea}(t)$ is a relaxation coefficient that varies over time, which constantly adjusts the model results during the integration process, making it approximate to the observed values while being constrained by the dynamical constraints of the physical model. The calculation process for $N_{rea}(t)$ is as follows:

$$N_{rea}(t) = \frac{\frac{\Delta t}{T}}{\frac{1+cos(2\pi \cdot \frac{t\%T}{T})}{T} + \Delta t} \tag{2}$$

$\Delta t$ is the time step in FAMIL2, which is 0.5h for C96 resolution (approximately 1-degree resolution). $T$ represents the time window with a value of 6 hours. As depicted in Figure 2a, the relaxation coefficient varies as a cosine function. It is large at the beginning and end of the temporal window, thereby facilitating accelerated convergence of the model results towards observations. While in the middle of the time window, $N_{rea}$ becomes smaller and even drops to zero, which indicates the reliability of the reanalysis data decreases. The reason is that the reanalysis data within the time window is obtained through interpolation between its start and end values.

In the forecast nudging, the initialization process adheres to a similar nudging algorithm at 6-h intervals, as shown in eq. (3).

$$x(t) = x_{model}(t) + N_{fcst}(t)[x_{fcst}(t) - x_{model}(t)] \tag{3}$$

Nevertheless, the atmospheric variables assimilated into the S2S system are sourced from the GFS weather forecast, denoted as $x_{fcst}(t)$. The relaxation coefficient $N_{fcst}(t)$ is as follows:

$$N_{fcst}(t) = \frac{\frac{\Delta t}{T}}{\frac{1+cos(2\pi \cdot \frac{t\%T}{T})}{T} + \Delta t} \cdot cos(\frac{\pi}{2} \cdot \frac{(t - t\%T)}{4mT}) \tag{4}$$

Compared to $N_{rea}$, $N_{fcst}$ is multiplied by a decay factor, which also varies in accordance with the cosine function. In this context, the number of days for forecast nudging is denoted by $m$, and the system is configured with a 10-day forecast nudging period. Figure 2b illustrates the variation of $N_{fcst}$, which decreases as the reliability of weather forecast data diminishes over time, ultimately reaching zero by the 10th day.



Summarily, the S2S forecast system commences its daily forecast from the initial condition derived via reanalysis nudging. It
then fine-tunes the forecasts with weather prediction data through the forecast nudging process. This initialization system
effectively reduces system errors in the model and augments forecast accuracy.

**2.3 Time-lagged method for ensemble generation**

The value of ensemble forecasts in medium to long-term forecasts has been repeatedly emphasized (Liu, 2003; Vitart and
Molteni, 2009). In addition to improving the physical scheme of the model, devising an effective approach for ensemble
generation might have a considerable impact on the MJO forecast. The IAP-CAS S2S ensemble forecast system utilizes the
time-lagged method (Hoffman and Kalnay, 1983) to generate ensemble members.
A schematic diagram of the time-lagged method is depicted in Figure 2b. During the initial day of the forecast nudging, the
S2S system issues forecasts from 00Z, 06Z, 12Z, and 18Z, resulting in the generation of 4 ensemble members. The core idea
behind this approach is to introduce perturbations by leveraging lagged initialization times.

**2.4 Hindcast experiment and real-time forecast**

The S2S ensemble forecast system provides daily forecasts, forecasting weather and climate conditions for the upcoming 65
days. Out of the 65 days, 5 days are reserved for extending the ensemble members by using the time-lagged method, ensuring
a complete forecast for at least 60 days. Since June 1st, 2019, the IAP-CAS S2S system has been operating 16 ensemble
members daily for real-time forecast, and for hindcast experiments from 1999 to 2018, it has run 4 ensemble members daily.
Our subsequent research is based on the 20-year hindcast experiment.
In 2021, the IAP-CAS model participated in phase II of the S2S Project (Vitart et al., 2017) successfully, providing the 20-
year hindcast and real-time forecast data generated by the S2S ensemble forecast system. Detailed information regarding the
data is listed in Table 4, and Table 5 shows the list of output variables. The output data is interpolated to a standardized
horizontal resolution of 1.5°×1.5°, following the S2S's requirements, and is stored in version 2 of General Regularly-distributed
Information in Binary (GRIB2) format. The output data of the S2S system is publicly available on three S2S Data Portals
(ECMWF, CMA, and IRI).

**3 Datasets and methods**

**3.1 datasets**

The observational datasets used for the MJO verification include the NOAA daily outgoing longwave radiation (OLR;
Liebmann and Smith 1996), daily wind from the National Centers for Environmental Prediction (NCEP)/Department of Energy
(DOE) Reanalysis 2 dataset (Kanamitsu et al., 2002), daily specific humidity from ECMWF Reanalysis version 5 (ERA5;
ERA 2017), and the precipitation product from the Global Precipitation Climatology Project (GPCP; Adler et al. 2003). To



facilitate computation and meaningful comparisons, both observation and hindcast datasets have been uniformly interpolated
to a horizontal resolution of 2.5°×2.5°. Seven pressure levels (1000, 925, 850, 700, 500, 300, and 200hPa) of wind and specific
humidity are extracted for analysis.
**3.2 MJO RMM index**
To conduct a quantitative assessment of MJO, we have employed the widely used Real-time Multivariate MJO (RMM) index
(Wheeler and Hendon, 2004a) to extract the MJO signal. This index consists of two components, RMM1 and RMM2, which
are the first and second principal components of the combined empirical orthogonal functions (EOFs) of multiple variables,
including OLR, 200hPa zonal wind (U200), and 850hPa zonal wind (U850). It serves as a tool for tracking the location and
amplitude characteristics of MJO.
The calculation of the RMM index refers to the method described in Gottschalck et al. (2010). Detailed calculation steps are
as follows:
1)   Remove the 0-3 waves of the climatology and low-frequency variability of the U200, U850, and OLR variables from both

195         the observation and hindcast data. It is noteworthy that removing low-frequency variability is to subtract the mean of the

196         past 120 days from the anomalies. For model forecast, this is the mean model anomalies of the previous forecast days,

197         plus the mean observed anomalies of the remaining days.

2)   Average the anomalies between 15° S and 15° N and normalize the three variables, using the pre-computed coefficients

199         as in Gottschalck et al. (2010).

3)   Project the anomalies onto the observed combined EOF eigenvectors from Wheeler and Hendon (2004b) to get RMM1

201         and RMM2.

Bivariate anomaly correlation coefficient (ACC) and bivariate root mean square error (RMSE) are calculated using the
observed and hindcast RMM indices to represent the forecast skills of the IAP-CAS model as
$$ACC(\tau) = \frac{\sum_{t=1}^{N}[a_1(t)b_1(t,\tau)+a_2(t)b_2(t,\tau)]}{\sqrt{\sum_{t=1}^{N}[a_1^2(t)+a_2^2(t)]}\sqrt{\sum_{t=1}^{N}[b_1^2(t,\tau)+b_2^2(t,\tau)]}}, \text{ and} \qquad (5)$$
$$RMSE(\tau) = \sqrt{\frac{1}{N}\sum_{t=1}^{N}[(a_1(t)-b_1(t,\tau))^2 + (a_2(t)-b_2(t,\tau))^2]} \qquad (6)$$
Here $a_1(t)$ and $a_2(t)$ are the observation RMM1 and RMM2 at time $t$; $b_1(t)$ and $b_2(t)$ are the forecasting RMM1 and
RMM2 at time $t$ for lead $\tau$ days; $N$ is the total number of times. It is commonly accepted that days with ACC above 0.5 are
considered to have valid forecasts. Therefore, the forecast skill of a model is quantitively defined as the maximum lead time
exceeding 0.5, which approximately corresponds to the day when RMSE reaches $\sqrt{2}$.
RMM index can also be adapted to quantitively evaluate the forecasted intensity and velocity through the calculation of the
error of amplitude ($ERR_{amp}(\tau)$) and phase ($ERR_{phase}(\tau)$) as a function of lead time $\tau$:
$$ERR_{amp}(\tau) = \frac{1}{N}\sum[AMP_b(t,\tau) - AMP_a(t)], \text{ and} \qquad (7)$$
$$ERR_{phase}(\tau) = \frac{1}{N}\sum tan^{-1}[\frac{a_1(t)b_2(t,\tau)-a_2(t)b_1(t,\tau)}{a_1(t)b_1(t,\tau)+a_2(t)b_2(t,\tau)}] \qquad (8)$$





Negative (positive) $ERR_{amp}(\tau)$ indicates weaker (stronger) amplitude in forecasts. Similarly, Negative (positive)
$ERR_{phase}(\tau)$ indicates slower (faster) propagation in forecasts. Here the MJO amplitude for observation ($AMP_a(t)$) and
forecast ($AMP_b(t)$) is defined as
$AMP_a(t) = \sqrt{a_1(t)^2 + a_2(t)^2}$, and                                                                                      (9)
$AMP_b(t,\tau) = \sqrt{b_1(t,\tau)^2 + b_2(t,\tau)^2}$.                                                                          (10)
**3.3 Cluster analysis of MJO events**
Another crucial method used in this research is the cluster analysis. In Section 5, we select the representative MJO events
and classify them following the work Wang et al. (2019) did. This facilitates a more focused and targeted investigation into
the forecast bias of MJO in the IAP-CAS model.
An MJO event was chosen if the regional average of OLR, spanning from 10° S to 10° N and 75° E to 95° E, remained
below one standard deviation for a consecutive period of 5 days during the boreal winter (November–April). Subsequently,
the K-means cluster analysis is employed to categorize the chosen MJO events based on the propagation patterns from day -
10 to 20 (day 0 is the day with the peak MJO in the Indian Ocean). At last, we use silhouette clustering evaluation criteria to
identify and eliminate poorly classified MJO events.
Finally, a total of 50 MJO events were selected from 1999 to 2018 winter and four types of MJO events were identified,
namely the fast-propagating (10 cases), slow-propagating (16 cases), standing (12 cases), and jumping (12 cases) patterns of
MJO (Fig. 5).
The fast-propagating MJO and slow-propagating MJO belong to the propagating type of MJO, characterized by their
consecutive eastward propagation across the Indian Ocean to the Pacific Ocean region. On the other hand, the standing and
jumping MJO represent relatively non-propagating types, where the convection remains relatively fixed or exhibits
inconsecutive movement. Wang et al. (2019) believe that propagating MJO events are often associated with strong and
tightly coupling Kelvin waves, especially for fast-propagating MJO. This is the biggest difference between propagating MJO
and non-propagating MJO.
**4 Evaluation of MJO forecast skill from the IAP-CAS model**
Figure 3 demonstrates the overall MJO forecast skill in the IAP-CAS model and the improvement brought by the time-lagged
ensemble method. Figure 3a shows the forecast skill of the ensemble mean is 24 days with the criterion of ACC exceeding 0.5,
while the skill of individual members is about 21-22 days. Meanwhile, the ensemble mean RMSE reaches $\sqrt{2}$ at 21 days and
the individual members exhibit larger RMSE, reaching $\sqrt{2}$ at 16 days (Fig. 3b). The solid blue line in Figure 3b represents the
ensemble spread (Leutbecher and Palmer, 2008) of IAP-CAS. When this ensemble spread approaches the RMSE of the
ensemble mean (solid red line), it indicates that the ensemble members are sufficiently dispersive. Figure 3b illustrates that the
ensemble exhibits an underdispersive characteristic in the early stage of the forecast. We have also observed similar issues of
" underdispersive" in many other models (Rashid et al., 2011; Neena et al., 2014; Kim et al., 2014b; Xiang et al., 2015), and
addressing this aspect may be a focal point for future model enhancements.
Increasing the number of ensemble members within a certain range proves effective in forecasting the uncertainty of weather
and climate (Hou et al. 2001). We employed the time-lagged ensemble method to further augment the ensemble members. The
time-lagged ensemble includes the ensemble members generated on the forecast day and from lag times. For instance, by
incorporating ensemble members with a lag of $i$ ($i = 0, 1, 2, ...$) days, the total number of members becomes $4 * (i + 1)$.
Upon examining the relationship between lag $i$ days and forecast skill, it was found that the skill increases as $i$ increases at
first, but then it reaches a plateau when $i > 3$ (see Fig. A3). This suggests that the forecast skill of the 16 members may
represent the limit of the time-lagged ensemble method in IAP-CAS. Figure 3d shows the ensemble of 16 members is more
dispersive than 4 members, which is illustrated by less distinction between RMSE and Spread in the 16-member system. The
ensemble mean of 16 members achieves a skill of 26 days, surpassing the skill of 4 members by two days (Fig. 3c).
Numerous prior investigations have demonstrated that MJO forecast skill is sensitive to the MJO amplitude in many models
(Lin et al., 2008; Rashid et al., 2011; Wang et al., 2014; Xiang et al., 2022), and this characteristic is also evident in the IAP-
CAS model. We classify an MJO case as an initial (target) strong case if its initial (target) amplitude is greater than 1, while
an event with an initial (target) amplitude less than 1 is classified as an initial (target) weak case. Figures 4a-b show that in the
IAP-CAS model, the forecast skills of strong MJO cases are generally higher than weak cases, especially in the target strong
(weak) cases.
The amplitude and phase of MJO serve as additional indicators for a detailed assessment of MJO forecast performance. For
initially strong MJO cases, we analyze the MJO amplitude and forecasted phase angle error (Figs. 4b-c). The individual
member has a stronger amplitude than observation, which leads to a relatively strong amplitude in the ensemble mean during
the initial 40 days. However, as the noise rapidly increases, the phase error of the individual members also escalates (as shown
in Fig. 4c). The phase error results in a mutual cancellation in positive and negative phases of MJO among ensemble members,
leading to a rapid weakening of the amplitude in the ensemble mean. In Figure 4d, the phase error of the ensemble mean
indicates that the speed of forecasted MJO tends to decrease at first and then start increasing around the $10^{th}$ day. A more
detailed investigation into the speed of propagating MJO events will be described in Section 5.
**5 The forecast of MJO propagation**
We present a qualitative diagnostic of a 20-year hindcast experiment to evaluate the overall forecast skills of IAP-CAS in
Section 4. This analysis provides us with preliminary insights into the performance and biases of the system. Based on Wang



et al. (2019), we aim to conduct further investigations into different types of MJO events to explore the physical explanation
of system biases.
In Section 3, we have already described the methodology for classifying MJO events and results. Figure 5 compares the
composited propagation patterns of precipitation and U850 between observation and forecast initiated at day -5 for four
different MJO types. The fast-propagating and slow-propagating MJO exhibit a consecutive eastward propagation structure
from the Indian Ocean across the MC region to the Pacific Ocean. The basic distinction between the two types is that the first
type of MJO exhibits a faster propagation speed compared to the second type. The standing MJO remains relatively stationary
over the Indian Ocean and does not continue to propagate eastward. The jumping MJO shows a convective system that
bypasses the MC region and directly jumps from the Indian Ocean to the Pacific Ocean. Here, fast MJO and slow MJO are
considered propagating MJO events, while the latter two types are regarded as non-propagating MJO events.
The observed U850 displays a coupled structure characterized by equatorial westerly anomalies of the Kelvin wave component
located west of the convection, and easterly anomalies of the Rossby wave component located east of the convection (Rui and
Wang, 1990b; Adames and Wallace, 2014; Wang and Lee, 2017). As illustrated in Figure 5, a distinct contrast between
propagating MJO and non-propagating MJO can be found in the circulation at the low level: in the propagating MJO events,
the Kelvin wave response is strong and tightly coupled with the center of convection, which is shown in the stronger and
eastward-extending easterly wind component, particularly prominent in fast MJO events. Many previous studies (Benedict
and Randall, 2007; Hsu and Li, 2012; Wang and Lee, 2017) have also indicated that the presence of low-level easterly winds
is a key signal that contributes to the eastward propagation of MJO by inducing low-level convergence and premoistening to
the east of the major convection. In the non-propagating MJO events, the easterly wind is weak and tends to decouple from
the major convection.
From the Hovmöller diagram of observed propagating MJO (Fig. 5), a significant change in convection is observed after
crossing the MC region, which is marked by a decrease in intensity and a slower propagation speed. This change is roughly
delineated by the 135° E, which is commonly referred to as the "MC barrier". As mentioned above, the "MC barrier" effect is
usually amplified in the climate models. This phenomenon is also observed in the forecast of slow MJO events in the IAP-
CAS model. The forecasted convective signal of slow MJO gradually fades after crossing the MC. However, in the forecast of
fast MJO, the convection does not exhibit a significant decrease in intensity after crossing the MC region, and the speed of the
propagation appears to be relatively faster compared to observations in both fast and slow MJO events. Figure 5 also shows
that the forecast for standing MJO remains somewhat imprecise. The forecasted convection is weak, and there are signals in
both the Indian Ocean and the Pacific Ocean, whereas observed standing MJO only indicates strong convective signals in the
Indian Ocean. This aspect is also evident in Figure A4, where the standing MJO has the lowest skill (13 days). For each MJO
type, we consider the skill as the ACC of the cases initiated from day -20 to day 15 (Xiang et al., 2015). Figure A4 displays




that the fast MJO achieves the highest skill at 32 days, while the jumping MJO and slow MJO exhibit skills of 23 and 21 days,
respectively.
In this work, we focus specifically on the analysis of propagating MJO events which have relatively complete propagation
processes. Figures 6 and 7 present the evolution patterns of propagating MJO. It is noticeable in both the spatial propagation
diagram (Fig. 6) and the phase diagram (Fig. 7) that the forecasted precipitation intensity is significantly higher than the
observed, indicating the presence of a stronger convective system of forecasted fast MJO. The forecasted location of the major
convection is relatively biased towards the east (Fig. 6b), which means there is an overestimation of the propagation speed.
The phase diagram also indicates a higher speed, with the blue points propagating faster than the red points (Fig. 7a). On the
15th day, the MJO convective system crosses the MC region and reaches the eastern Pacific (Figs. 6a-b). It is worth noting
that the forecasted negative phase of MJO exhibits a significant development, with an accelerated speed, gradually intruding
into the positive phase. By the 20th day, the development of the negative phase has further intensified, extending its influence
into the tropical eastern Pacific region, while in the observation, the negative phase remains east of the MC region. A similar
phenomenon is also evident in the forecasts of slow MJO events. Figures 6c-d and 7b show that the amplitude of slow MJO is
weaker than fast MJO. There are still biases of stronger convection and faster propagation in the forecast of slow MJO. In the
later stages, as the negative phase intrudes, the forecasted convective signal in the positive phase is almost absent due to the
inherently weaker convection in slow MJO. This amplification of the "MC barrier" in the forecast of slow MJO may contribute
to the diminished convective signal. The intrusion of negative-phase convection is also observed in forecasted fast MJO.
However, due to the relatively strong positive-phase convection in the forecast of fast MJO, it is less evident in the zonal mean,
as shown in the Hovmöller diagram.

## 6 The possible physical explanation for the forecast biases

Section 5 highlights some biases observed in the forecast of propagating MJO, which includes stronger amplitude and faster
propagation speed in the IAP-CAS model. These biases are also mentioned in Section 4. In this section, we aim to unravel the
physical mechanisms underlying these biases.
As a large-scale convective system, MJO's genesis, evolution, and dissipation are intricately linked to atmospheric moisture
(Wang, 1988; Kemball-Cook and Weare, 2001; Maloney, 2002; Wang and Lee, 2017). Given that the model forecasts exhibit
a systematic bias of stronger amplitude, we start with the diagnosis of the background state in moisture. Figure 8 shows the
winter mean specific humidity averaged over 10° S–10° N. A clear positive bias of the background moisture state in the IAP-
CAS model is observed (Fig. 8c), which can enhance the convection in the MJO. However, the distribution of this moisture
bias is non-uniform. Figure 8c illustrates that the positive moisture bias is more pronounced towards the western Indian Ocean
and the central-eastern Pacific, and this bias gradually spreads to the upper levels. However, in the MC region, the positive



moisture bias is smaller and primarily concentrated in the low level. We speculate that higher evaporation fluxes in the model
may be the reason for the positive moisture bias. Furthermore, the reduction in oceanic surface area within the MC region
contributes to a decrease in this positive bias.
Figure 9 displays the precipitation-induced condensational heating ($Q_2$) during fast MJO and slow MJO events. The
condensational heating serves as a proxy for the distribution of convection, which was estimated by the moisture sink defined
as
$Q_2 = -L_v(\frac{\partial q}{\partial t} + \vec{V} \cdot \nabla q + \omega \frac{\partial q}{\partial p})$,    (11)
where $q$ is the specific humidity, $\vec{V}$ is the horizontal circulation, $\omega$ is vertical pressure velocity, and $L_v$ is the latent heat
at condensation, which is a constant here. The vertical distribution of $Q_2$ reveals that both fast MJO and slow MJO events in
the model forecasts trigger stronger convection, particularly in the fast MJO events. Furthermore, the enhanced convective
heating leads to a strong response in the coupled MJO-related circulation (Fig. 9). On the 10[th] day, both the fast MJO and the
slow MJO experience further intensification in the model. This intensification can be attributed to the amplification of the
positive moisture bias following the departure from the MC region.
To further understand the faster propagation of MJO in the IAP-CAS model, it is necessary to comprehend the underlying
physical processes associated with the propagation of MJO. Under the framework of "moisture mode", Jiang (2017) performed
a moisture budget analysis on the latest generation of general circulation models (GCMs) and identified the key processes for
the eastward propagation of MJO. This research revealed that the advection ($\overrightarrow{V'} \cdot \nabla \bar{Q}$) of the seasonal mean moisture ($\bar{Q}$) by
the MJO anomalous circulations ($\overrightarrow{V'}$) plays a crucial role in the propagation of MJO. By increasing moisture eastward and
decreasing it westward of the MJO convection, the advection regulates the propagation. (Kim et al., 2014a; Adames and Kim,
2016; Jiang et al., 2018). Among the two determining factors ($\overrightarrow{V'}$ and $\bar{Q}$), the role of the moisture gradient term is further
emphasized. Many studies (Gonzalez and Jiang, 2017; DeMott et al., 2018; Ahn et al., 2020) have demonstrated that the mean
moisture's horizontal gradient plays a crucial role in determining the propagation of MJO (Fig. 10a). It is well-forecasted in
the models that simulate MJO well, leading to realistic horizontal mean moisture gradients and, thus, well-forecasted horizontal
moisture advection associated with the MJO (Hsu and Li, 2012; Kim et al., 2014a; Nasuno et al., 2015; Adames and Wallace,
2015; Gonzalez and Jiang, 2017). The IAP-CAS model is capable of reproducing this gradient, although there is an overall
stronger moisture bias (Fig. 10b). Here, the $\bar{Q}$ presented is the winter mean specific humidity at 850 hPa ($\bar{Q}_{850}$). Research has
indicated that the $\bar{Q}_{850}$ is representative (Kim, 2019), and subsequent analysis also focuses on the 850 hPa level.
Figure 11 shows the composite equatorial U850 averaged over the 15° S-15° N for fast MJO and slow MJO respectively. It
depicts the transition from westerly to easterly winds in the MC region (as enclosed by the two blue dashed lines), leading to
the change from positive advection to negative advection. On the 1[st] and 5[th] days, the MC region is predominantly occupied
by easterly winds, while from the 10[th] to the 20[th] day, the region is primarily characterized by westerly winds in both fast MJO



and slow MJO. However, the forecasted amplitude of low-level wind is significantly stronger, which can be caused by the
enhanced MJO convection as explained earlier.
The MJO anomalous circulation patterns in the MC region result in a positive moisture advection on the eastern part of the
MJO during the early stages of MJO's development, which facilitates the propagation of convection in the positive phase. We
refer to this process as the "developing phase". Figure 12 provides a detailed illustration of this process. Conversely, during
the later stages, there is a negative moisture advection on the western side of the MJO, which leads to the propagation of
convection in the negative phase and promotes the dissipation of the MJO. We refer to this process as the "decaying phase"
(Fig. 12). Compared to the observation, the stronger amplitude of the low-level moisture advection ($\vec{V'} \cdot \nabla \bar{Q}$) in the model
explains the accelerated positive phase of convection during the early stages and the accelerated negative phase during the
later stages (Fig. 13). This explains the increasing propagation speed of the forecasted MJO. In the observation, the amplitude
of the moisture advection during fast MJO events is stronger than that during slow MJO events, further confirming this physical
explanation.
In addition to examine the winter mean moisture state ($\bar{Q}$), we have analyzed MJO-related moisture anomalies ($Q'$) as well
(Fig. 14). By comparing the evolution pattern of moisture anomalies between slow MJO and fast MJO, it is found that the
moisture anomalies in the eastern part of fast MJO are more intense compared to the slow MJO. This results in stronger low-
level moisture transport towards the convective region, thereby also facilitating the intensification and acceleration of the MJO.
Moreover, there is a significant distinction in the spatial correlation between fast and slow MJO and it happens as early as the
1$^{st}$ day. As the forecast lead time progresses, the accuracy of the moisture forecast deteriorates, while fast MJO events display
comparatively better performance. The disparity in moisture anomalies is possibly a pivotal factor contributing to differences
in forecast skills between the fast (32 days) and the slow MJO (21 days). This underscores the significance of improving
moisture forecast in the MJO forecast.
**7 Summary and discussion**
**7.1 Summary**
The graphical abstract presents a workflow for this paper, outlining the structure of this work. This study introduces a newly
developed S2S ensemble forecast system of the IAP-CAS model. The introduction primarily focuses on the numerical model,
initialization, ensemble generation, and post-processing aspects of the S2S system. Then we aim to identify potential
possibilities for developing this S2S system through a comprehensive assessment of its forecast skills. Based on the 20-year
hindcast experiment, the IAP-CAS model shows comparable skill (24 days) to other S2S models. However, the ensemble
forecast for MJO has been demonstrated to be underdispersive. A detailed examination of the propagating MJO forecasted in





the IAP-CAS model reveals that the amplitude of the convection is overestimated with an increasing propagation speed,
particularly in the fast MJO events. These biases are accompanied by a faster dissipation of the MJO.
The root cause of these biases lies in the model's wetter environment, which leads to enhanced convection and strengthened
circulation coupled with convection, and subsequently, stronger moisture convection. The increasing propagation speed in the
MJO propagation is mainly associated with the stronger amplitude of the low-level moisture advection ($\overrightarrow{V'} \cdot \nabla \bar{Q}$) in the forecast.
Moreover, the differences in forecast skills between the fast MJO and the slow MJO may be attributed to discrepancies in
moisture anomalies ($Q'$) forecast. This further underscores the significance of accurate moisture forecasts.
**7.2 Discussion**
In Figs A5, we compare the forecast skill of the IAP-CA model with ten other S2S models. The MJO index of 11 S2S models
and ERA-Interim from the S2S website (http://www.s2sprediction.net/) is used for evaluation during the standard hindcast
period 1999-2010. Among the 11 S2S models, the IAP-CAS model exhibits MJO skill above the mean skill level, while the
ECMWF model stands out as the highest-performing model. Figure A5b shows that the skill of individual members in ECMWF
is approximately 17-18 days, whereas the ensemble mean demonstrates an extended skill of up to 30 days. This phenomenon
may be attributed to the ECMWF model's considerable dispersion (Fig. A5c), which once again underscores the critical role
of ensemble dispersion in forecasting uncertainties of weather and climate.
Therefore, the forthcoming phase in our model's development plan encompasses increasing model dispersion through
improved ensemble perturbation methods, with the ultimate goal of improving model forecast skills. The method of orthogonal
conditional nonlinear optimal perturbations (CNOPs, Mu et al. 2003) and the second-order exact sampling (Pham, 2001) are
both promising approaches for generating initial perturbations in the model. This method allows the generation of a set of
initial perturbations in different orthogonal perturbation subspaces, each with the maximum potential for nonlinear
development. When applied to ensemble forecast using a simple Lorenz-96 model, the CNOPs method has demonstrated
higher forecast skill compared to the commonly used linear Singular Vectors (SVs) method (Lorenz, 1996). Furthermore,
PDAF (Parallel Data Assimilation Framework, Nerger et al., 2020) provides an efficient method known as second-order exact
sampling, which uses the long-time variability of the model dynamics to estimate the uncertainty. Evidence has already
suggested that the use of second-order exact sampling can greatly improve the skill in sea ice extent throughout the Arctic and
along the Northern Sea Route (Yang et al., 2020). We plan to explore the application of CNOPs and second-order exact
sampling in the IAP-CAS model in the future and eagerly anticipate the potentially significant results it may yield. Additionally,
using machine learning to improve the skill of ensemble forecast is also a viable way to enhance the ensemble forecast of our
model.
In addition to exploring ensemble perturbations, we also intend to enhance the initialization system of the model. Recognizing
the moisture is crucial in the forecast of MJO and acknowledging the issue of moisture bias in the IAP-CAS model, it is



essential to take measures to ameliorate moisture forecast in our model. The recent research by Zeng (Zeng et al., 2023)
provides convincing evidence that humidity initialization can indeed significantly enhance MJO forecast in the IAP-CAS S2S
forecast system, especially in the 2 and 3 phase of MJO propagation. However, it is worth noting that changes in the mean
state have a significant impact on MJO development (Hannah et al., 2015; Kim, 2019), we must pay attention to the influence
of moisture initialization on the mean state. Moreover, the current S2S system's initialization process uses the nudging method,
and it is worthwhile to explore more efficient methods to enhance the initialization process.
We are also considering increasing the resolution of the model to C384 (25 km) globally. A High-resolution coupled model
could better represent the MJO (Crueger et al., 2013). This improvement could be attributed to the enhanced resolution, which
better captures the ocean-atmosphere interaction – a critical factor for MJO convection. Increasing the resolution is also
meaningful for addressing the MC barrier issue because one of the factors contributing to the MC barrier is terrain distortion
(Hsu and Lee, 2005; Inness and Slingo, 2006; Wu and Hsu, 2009). Further optimizing the model's physical processes and
dynamic-physical coupling is also believed to enhance the forecast of the MJO (Zhou and Harris, 2022). As the foreseeable
resolution and complexity of the model increase in the future, the issue of power consumption on X86 architecture processors
for handling the growing amount of data will become more pronounced. We have plans to port the model to the computing
platform based on ARM architecture to address the challenges posed by the explosive growth of data.



**Table 1 Configuration of the coupled climate system model CAS FGOALS-f2**

| Component | Model name | Horizontal Resolution | Vertical levels | Reference |
|---|---|---|---|---|
| Atmosphere | FAMIL2 | Cubed Sphere Grid (C96, ~1°×1°) | 32 in the hybrid levels | Li et al. 2019 |
| Land | CLM4.0 | Nested subgrid hierarchy (f09, ~1°×1°) | 15 soil levels and 3 snow levels | Oleson et al. 2010; Lawrence et al. 2011 |
| Ocean | POP2 | Displaced-pole grid (gx1v6, ~1°×1°) | 60 levels | Kerbyson and Jones 2005 |
| Sea ice | CICE4 | Displaced-pole grid (gx1v6, ~1°×1°) | 5 levels | Hunke et al. 2010 |



**Table 2 Hybrid coefficient of hybrid sigma-pressure coordinates at layer interfaces in CAS FGOALS-f2**

| Layer | Coefficient of pressure coordinates | The coefficient of sigma coordinates | Layer | Coefficient of pressure coordinates | The coefficient of sigma coordinates |
|---|---|---|---|---|---|
| 1 | 100.00 | 0.00 | 18 | 27131.33 | 0.23 |
| 2 | 400.00 | 0.00 | 19 | 24406.11 | 0.32 |
| 3 | 818.60 | 0.00 | 20 | 21326.05 | 0.42 |
| 4 | 1378.89 | 0.00 | 21 | 18221.18 | 0.51 |
| 5 | 2091.80 | 0.00 | 22 | 15275.15 | 0.59 |
| 6 | 2983.64 | 0.00 | 23 | 12581.68 | 0.67 |
| 7 | 4121.79 | 0.00 | 24 | 10181.43 | 0.73 |
| 8 | 5579.22 | 0.00 | 25 | 8081.90 | 0.79 |
| 9 | 7419.79 | 0.00 | 26 | 6270.87 | 0.83 |
| 10 | 9704.83 | 0.00 | 27 | 4725.35 | 0.87 |
| 11 | 12496.34 | 0.00 | 28 | 3417.39 | 0.91 |
| 12 | 15855.26 | 0.00 | 29 | 2317.75 | 0.93 |
| 13 | 19839.62 | 0.00 | 30 | 1398.09 | 0.96 |
| 14 | 24502.73 | 0.00 | 31 | 632.50 | 0.98 |
| 15 | 28177.10 | 0.02 | 32 | 0.00 | 0.99 |
| 16 | 29525.28 | 0.06 | 33 | 0.00 | 1.00 |
| 17 | 29016.34 | 0.14 | | | |



**Table 3 Initialization information of the S2S ensemble forecast system**

| Nudging type | Data Assimilation | Variable | Data | Frequency |
|---|---|---|---|---|
| Reanalysis nudging | Time-Lagged Nudging (Hoffman and Kalnay, 1983; Jeuken et al., 1996) | U, V, T, $P_s$, $z_s$ [a] | FNL (http://rda.ucar.edu/datasets/ds083.2, ds083.2\|DOI: 10.5065/D6M043C6) | 6h |
| | | SST | NOAA OISST (Reynolds et al., 2007) | |
| Forecast nudging | | U, V, T, $P_s$, $z_s$ | GFS weather forecast | 6h |

[a] Table notes: U represents zonal wind, V represents meridional wind, T represents temperature, $P_s$ represents surface pressure,
$z_s$ represents surface geopotential height, and SST represents sea surface temperature.



**Table 4 Introduction to the output data of the S2S ensemble forecast system**

| Experiment | Ensemble members | Time range | Frequency | Forecast time | Variable | Resolution | Interpolation method |
|---|---|---|---|---|---|---|---|
| Hindcast | 4 | 1999-2018 | Daily | 65 days | 25 variables (A detailed list of variables is shown in Table 5) | Horizontal:1.5° ×1.5° Vertical:7 levels (1000, 925, 850, 700, 500, 300, and 200hPa) | One-order conservation |
| Real-time forecast | 16 | 2019 | | | | | |



**Table 5 List the output variables in the S2S ensemble forecast system**

| Statistical process | Level(s) | Short name | Standard name | Unit |
|---|---|---|---|---|
| Instantaneous value/24h | The variables are located on 10 pressure layers (1000, 925, 850, 700, 500, 300, 200, 100, 50, 10 hPa) | gh | Geopotential height | gpm |
| | | t | Temperature | K |
| | | u | U-velocity | m s$^{-1}$ |
| | | v | V-velocity | m s$^{-1}$ |
| | | w | Vertical velocity | pa s$^{-1}$ |
| | The variable is located on 7 pressure layers (1000, 925, 850, 700, 500, 300, 200 hPa) | q | Specific humidity | kg kg$^{-1}$ |
| | 2-dimensional variables | w | Vertical velocity | pa s$^{-1}$ |
| | | sp | Surface pressure | Pa |
| | | lsm | Land sea mask | Proportion of land |
| | | orog | Orography | gpm |
| Daily average value | | tcc | Total cloud cover | % |
| | | skt | Skin temperature | K |
| | | 2t | Surface air temperature | K |
| | | 2d | Surface air dewpoint temperature | 2d |
| | | wtmp | Sea surface temperature | K |
| | | ci | Sea ice cover | proportion |



| 24-hour accumulated value | sf | Snow fall water equivalent | kg m$^{-2}$ |
|---|---|---|---|
| | ttr | Time-integrated top net thermal radiation | W m$^{-2}$ s |
| | ssr | Time-integrated surface net solar radiation | W m$^{-2}$ s |
| | str | Time-integrated surface net thermal radiation | W m$^{-2}$ s |
| | ssrd | Time-integrated surface solar radiation downwards | W m$^{-2}$ s |
| | strd | Time-integrated surface thermal radiation downwards | W m$^{-2}$ s |
| Instantaneous value/6h | mx2t6 | Surface air maximum temperature | K |
| | mn2t6 | Surface air minimum temperature | K |
| | 10u | 10 metre u-velocity | m s$^{-1}$ |
| | 10v | 10 metre v-velocity | m s$^{-1}$ |
| 6-hour accumulated value | tp | Total precipitation | kg m$^{-2}$ |



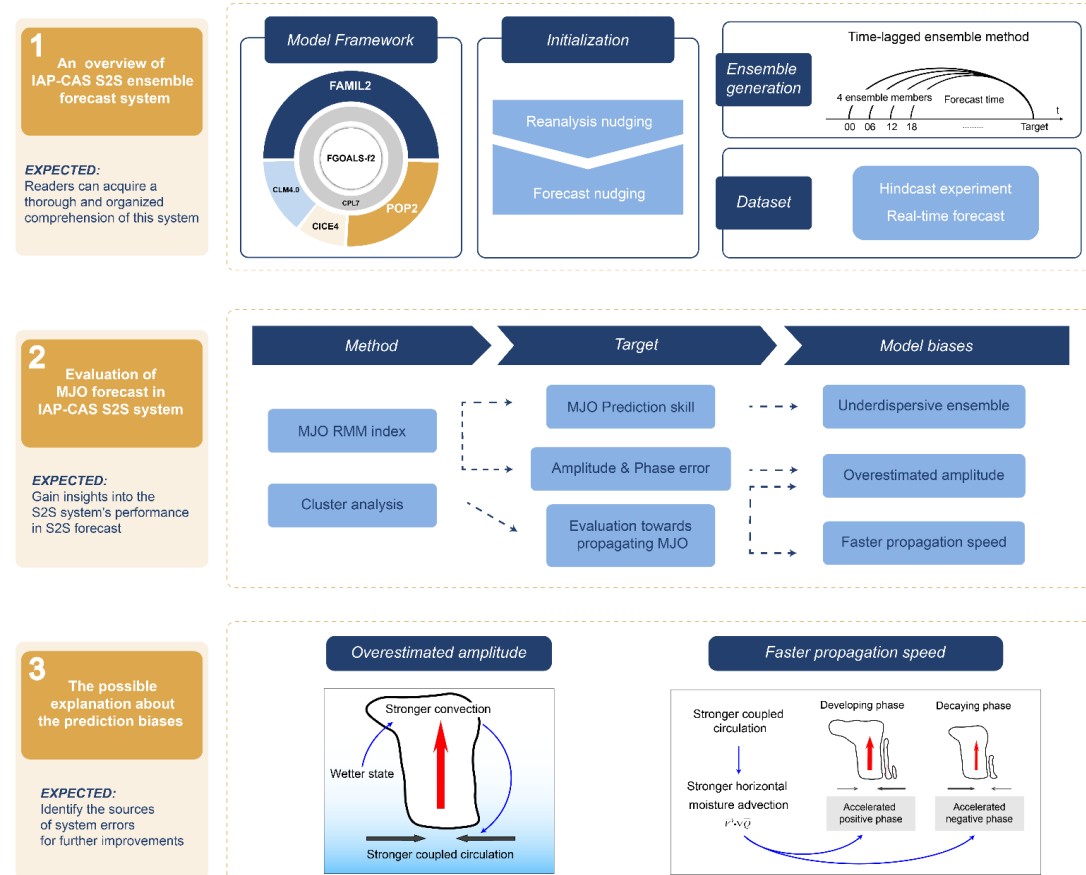

**The graphical abstract**





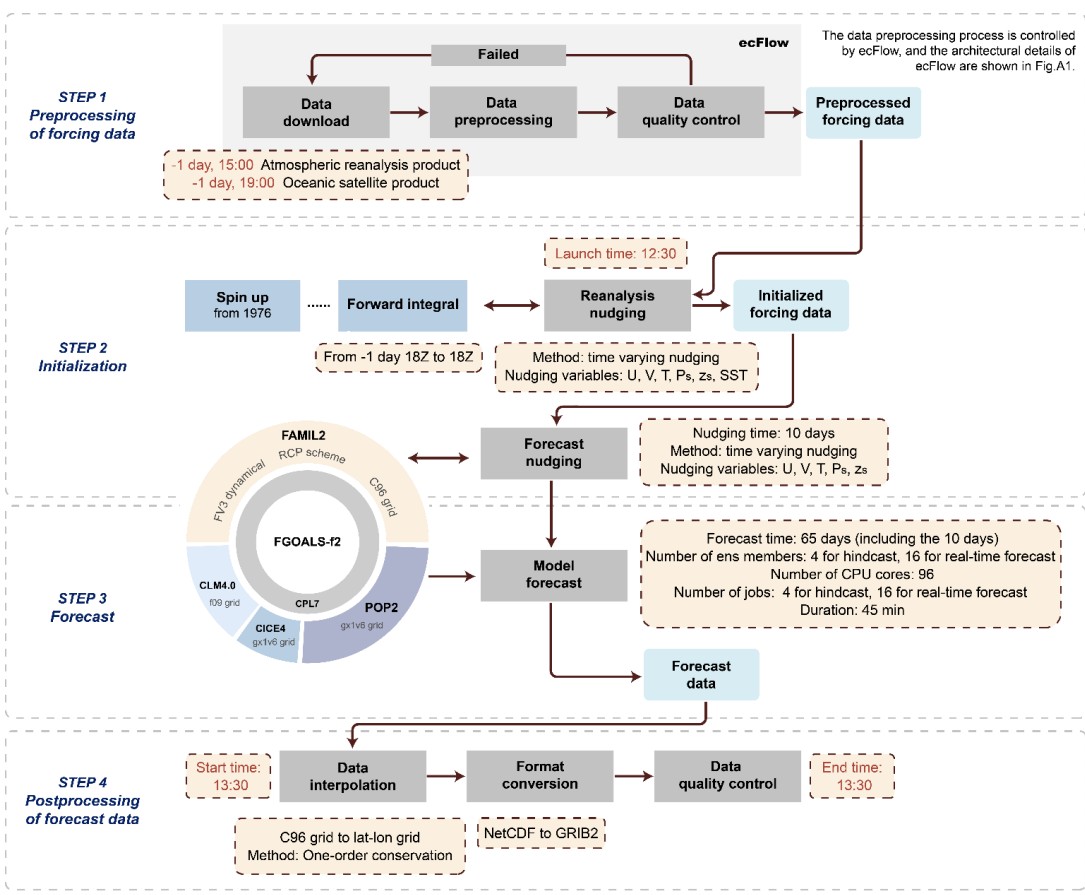

The time marked in red here is local time (beijing time)
**Figure 1. The structure of the IAP-CAS S2S ensemble forecast system**



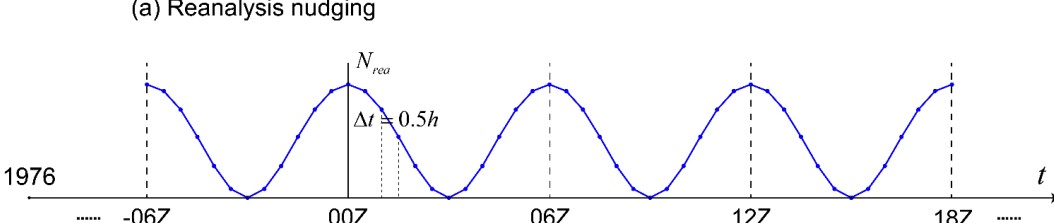

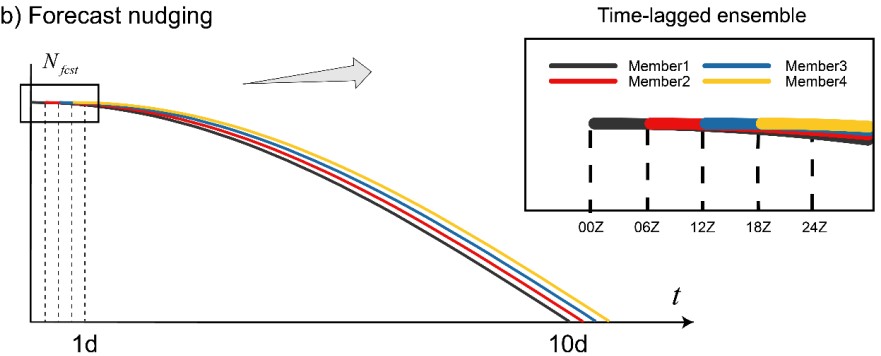

**Figure 2. The initialization scheme of the S2S ensemble forecast system in the IAP-CAS model. The relaxation coefficient (*N*) as a function of time (*t*) in (a) the reanalysis nudging and (b) the forecast nudging. In (a), The reanalysis nudging begins on January 1, 1976. The dots indicate the nudging process every 30 minutes. In (b), the solid lines of 4 colors represent the 4 ensemble members with their generation facilitated through the application of the time-lagged method.**



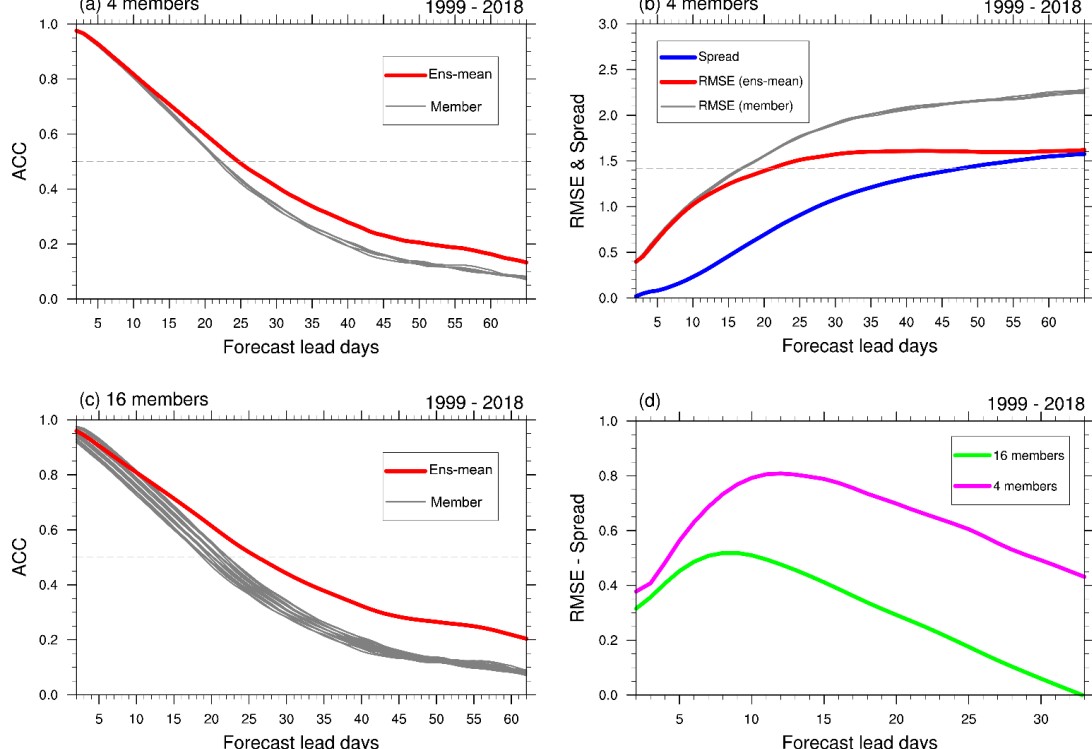

**Figure 3. MJO forecast skill of IAP-CAS over 20 years (1999-2018) re-forecast data. (a) The bivariate anomalous correlation coefficient (ACC) and (b) The Root Mean Squared Error (RMSE) varied with forecast lead days for individual members (gray solid line) and ensemble mean (red solid line). The blue solid line denotes the ensemble spread. (c) The ACC of individual members and ensemble mean. The dashed line in (a) and (c) has the values of 0.5, and it represents 1.414 in (b). (d) The difference between RMSE and Spread of 4-member ensemble mean (purple solid line) and 16-member ensemble mean (green solid line).**

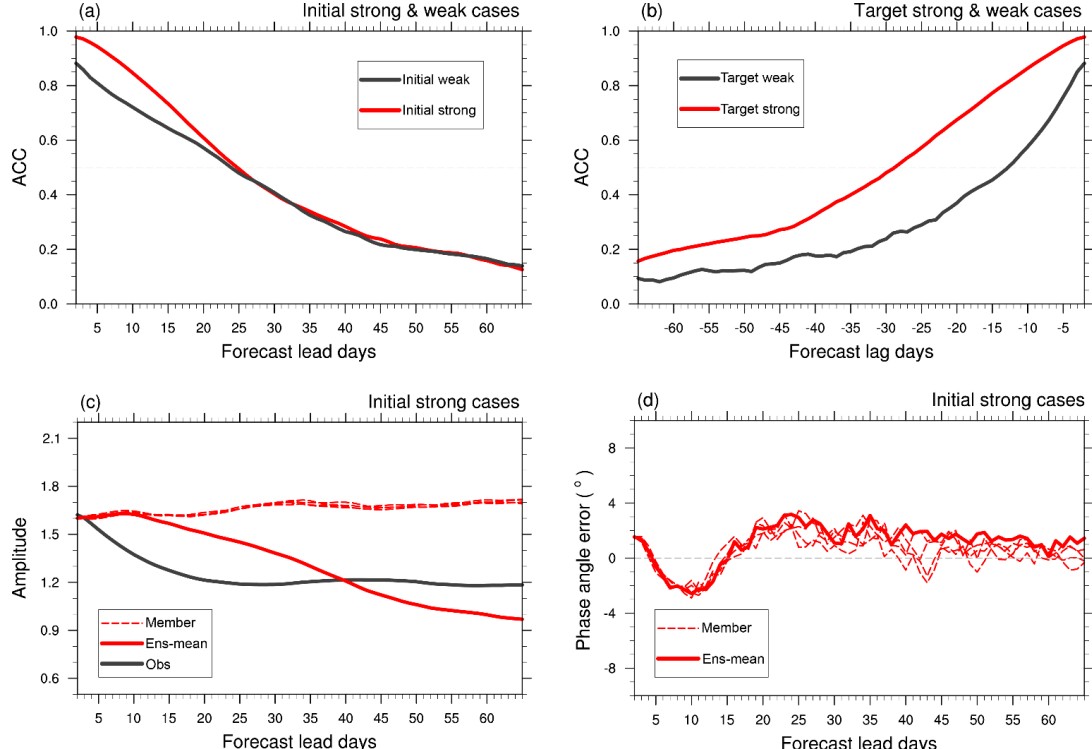

**Figure 4. The ACC (a) varied with forecast lead days for initially strong (red) and weak (black) cases and (b) varied with forecast lag days for target strong (red) and weak (black) cases from the ensemble mean. The dashed lines in (a) and (b) have the values of 0.5. (c) The forecast of MJO amplitude varied with forecast lead days for initially strong cases from observation (black solid line), individual ensemble members of the model (red dashed line) and their ensemble mean (red solid line). (d) The forecast of MJO phase angle error (°) for initially strong cases (black solid line). The dashed line in (d) is the reference line with the values of 0.**

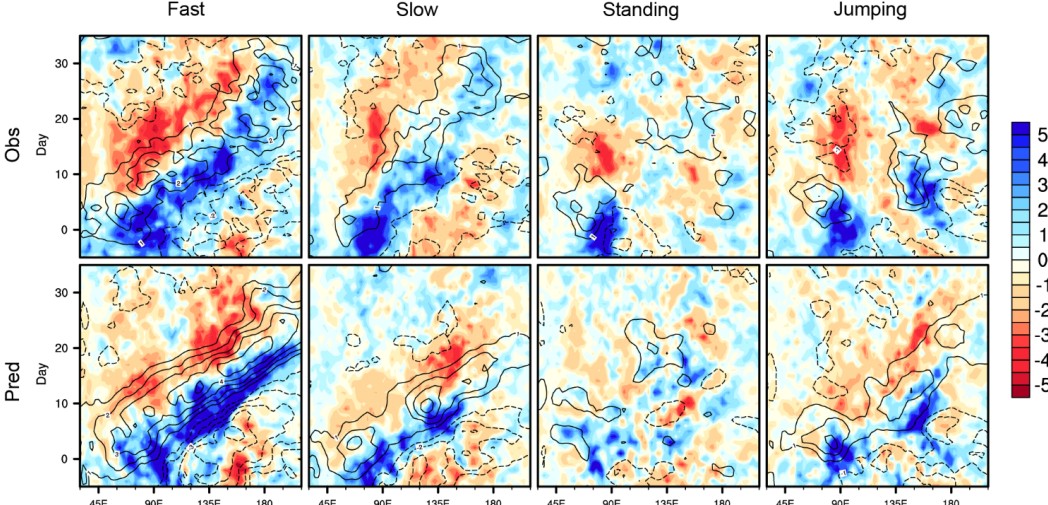

**Figure 5. 10° S–10° N averaged Precipitation anomalies (shading; mm day⁻¹) and 850-hPa zonal winds anomalies (contours with an interval of 1 m s⁻¹) varied with longitude (x-axis) and time lag (y-axis; days) composited for four types of MJO. The top row is for observation (NCEP winds and GPCP precipitation), and the bottom row is for model forecasts initiated at day −5 (5 days before the peak day). Solid lines represent positive values and dashed lines represent negative values.**



**Figure 6. Evolution patterns of the composite precipitation (shading; mm day⁻¹) and 850-hPa winds (vectors; m s⁻¹) anomalies for day 1, day 5, day10, day15 and day 20 in (a) observed fast MJO, (b) simulated fast MJO, (c) observed slow MJO and (d) simulated slow MJO.**



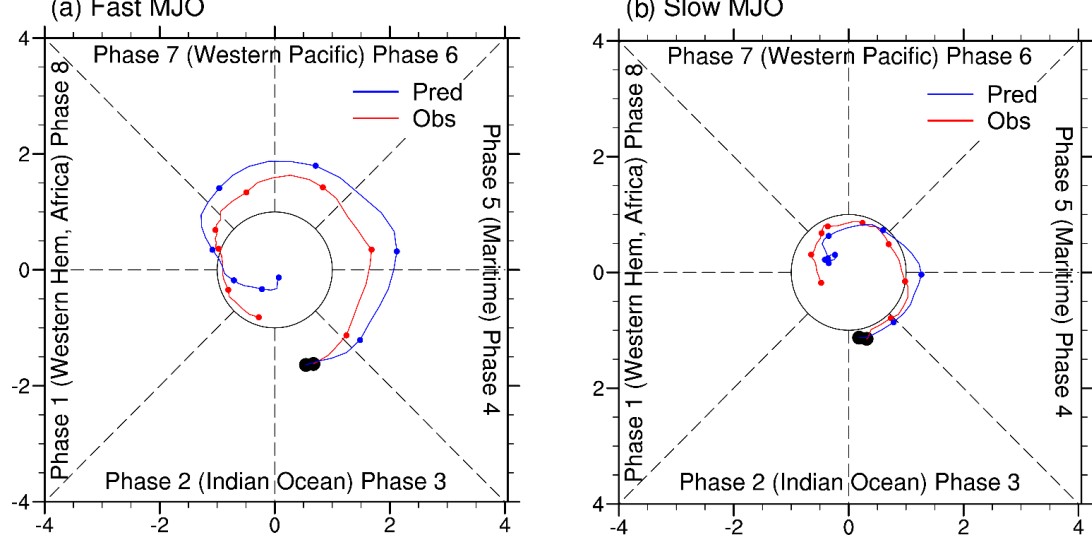

**Figure 7. Composite phase diagrams for (c) fast MJO and (f) slow MJO events from observation (red lines) and IAP-CAS model (blue lines). The dots denote every 5 days from the forecast starting date.**



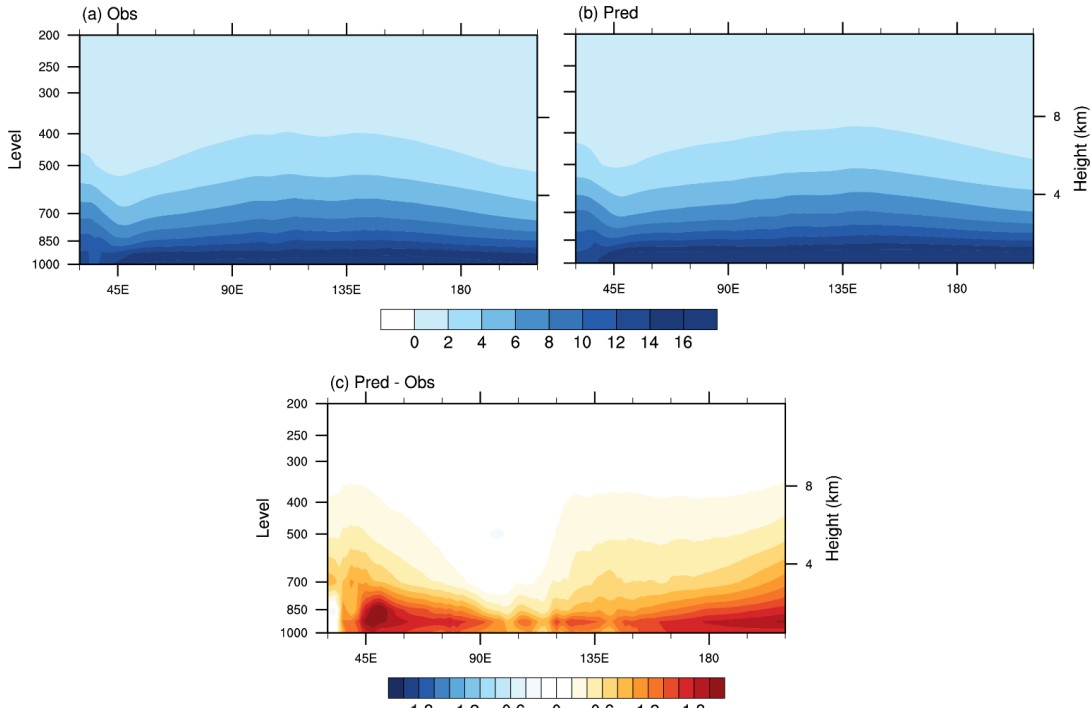

**Figure 8. The longitude-vertical profiles of winter (November–April) mean specific humidity (g kg⁻¹) averaged over 10° S–10° N for (a) observation, (b) IAP-CAS model, and (c) the difference between IAP-CAS model and observation.**

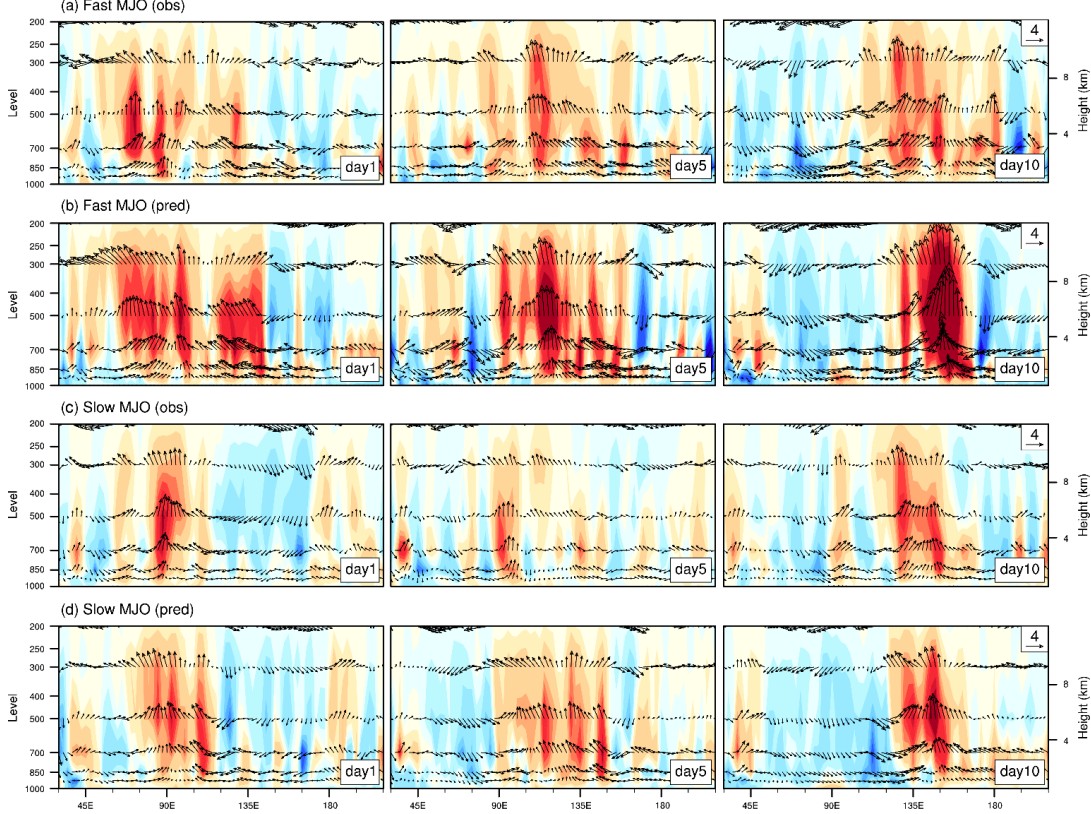

**Figure 9. The composited longitude-vertical structure of precipitation heating (contours; 1×10$^{-2}$ J kg$^{-1}$ s$^{-1}$) and zonal and vertical winds anomalies (vectors; units are m/s for zonal winds and 0.01 Pa s$^{-1}$ for vertical winds) averaged over 10° S–10° N for day 1, day 5, day 10 in (a) observed fast MJO, (b) simulated fast MJO, (c) observed slow MJO and (b) simulated slow MJO.**





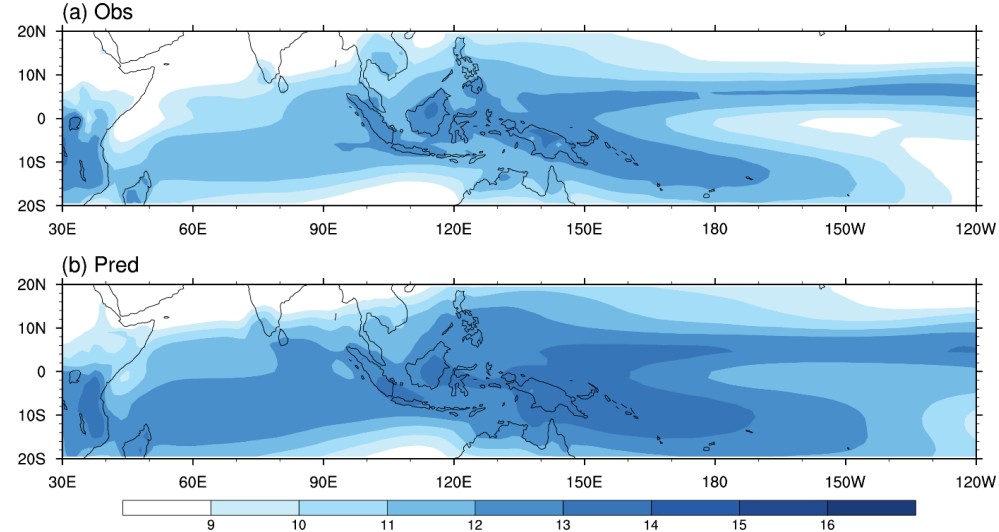

487

488    **Figure 10. The winter (November–April) mean specific humidity (g kg⁻¹) on 850hPa for (a) observation and (b) IAP-CAS model.**

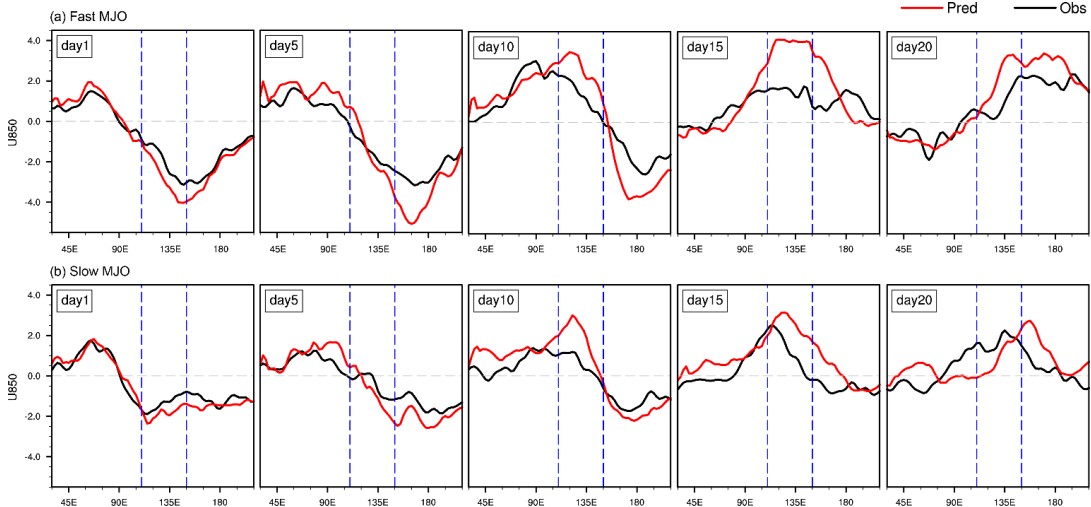

**Figure 11. The composited longitudinal structure of the 850hPa zonal wind anomalies (m s⁻¹) averaged over 15° S–15° N for day 1, day 5, day10, day15 and day 20 from observation (black solid line) and IAP-CAS model (red solid line) in fast and slow MJO events. The gray dashed line is the reference line with the values of 0. The two blue dashed lines are 110° E and 150° E respectively, which denote the extension of the MC region.**



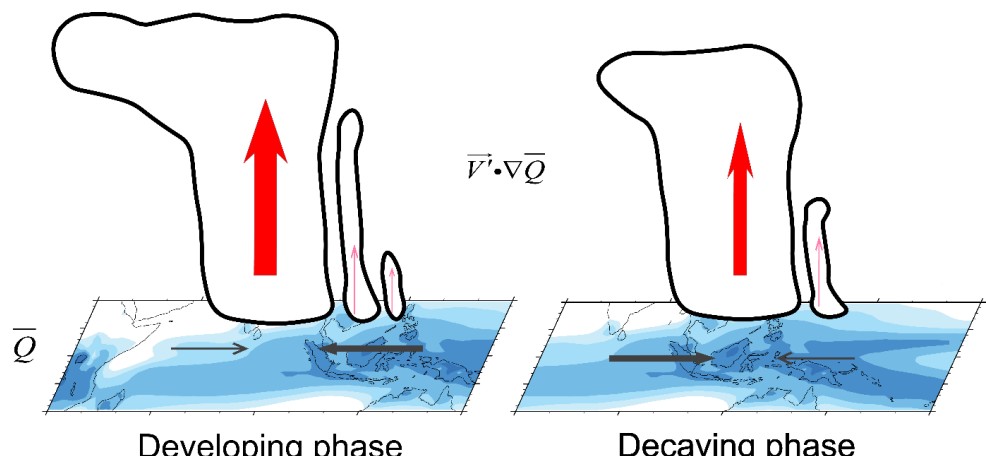

**Figure 12. Schematic diagrams illustrating the moisture mode theory on MJO propagation in the MC region.**



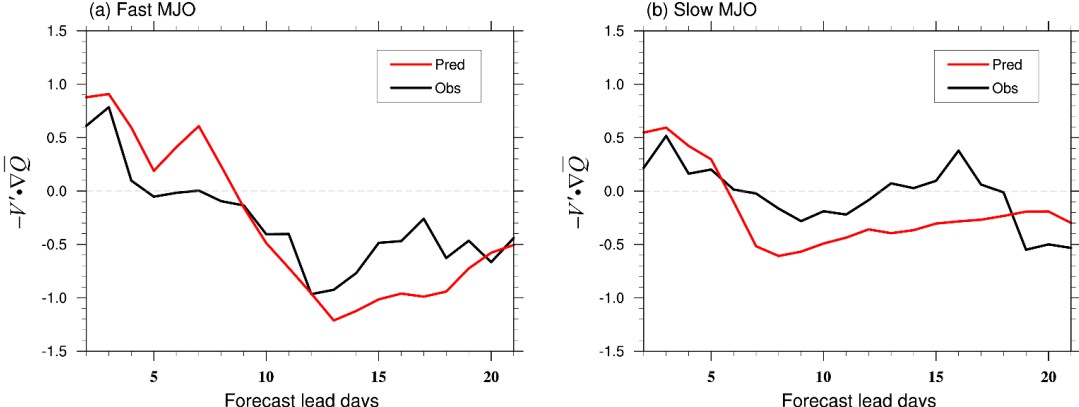


**Figure 13. The composited** $-V' \cdot \nabla \overline{Q}$ **(g kg⁻¹ s⁻¹) averaged over the MC region (15° S-15° N, 110° E-150° E) as a function of forecast**
**lead days from observation (black solid line) and IAP-CAS model (red solid line) in (a) fast MJO and (b) slow MJO events. The gray**
**dashed line is the reference line with the values of 0.**

**Figure. 14. Evolution patterns of the composite specific humidity anomalies (g kg⁻¹) and winds (vectors; m s⁻¹) anomalies on 850hPa for day 1, day 5, day10, day15 and day 20 (a) observed fast MJO, (b) simulated fast MJO, (c) observed slow MJO and (b) simulated slow MJO. The spatial correlation coefficient between simulated and observed moisture anomalies is shown to the right of panels (b) and (c).**



**Appendix**

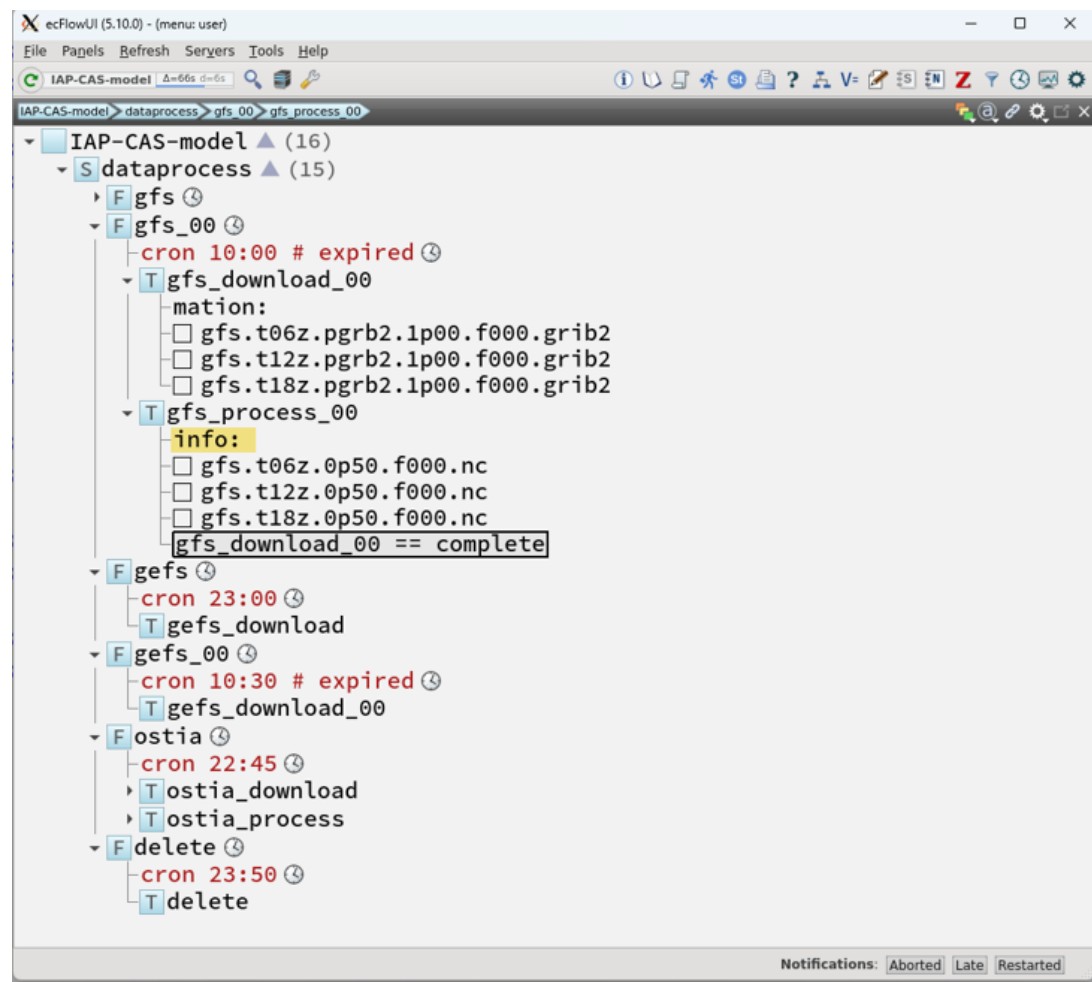

**Figure A1. The structure of ecFlow (ECMWF Workflow). EcFlow, developed and maintained by the ECMWF, is a client/server**
**workflow package designed to facilitate the execution of a substantial number of programs within a controlled environment. It is**
**used in the IAP-CAS model to accomplish the download and preprocessing of the forcing data.**



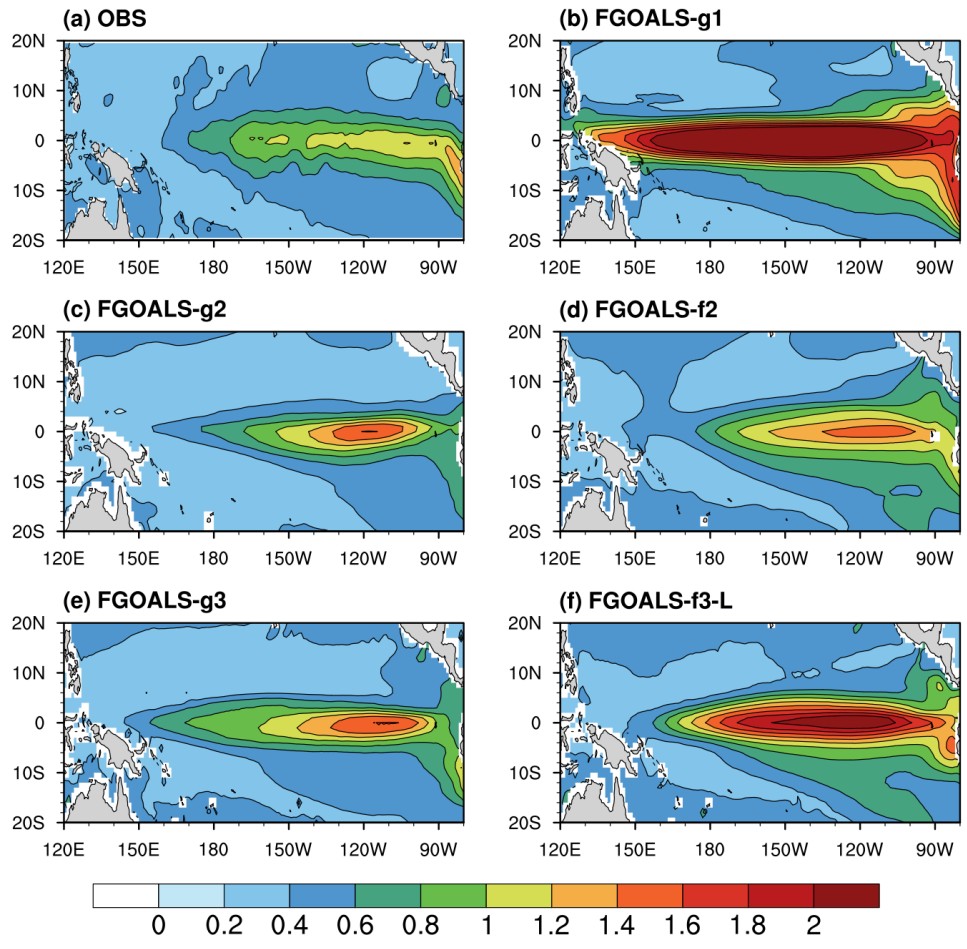

**Figure A2. The horizontal distribution of Sea Surface Temperature Anomaly (SSTA) standard deviations in (a) observation and (b)–(e) five FGOALS models from 1948 to 2018.**



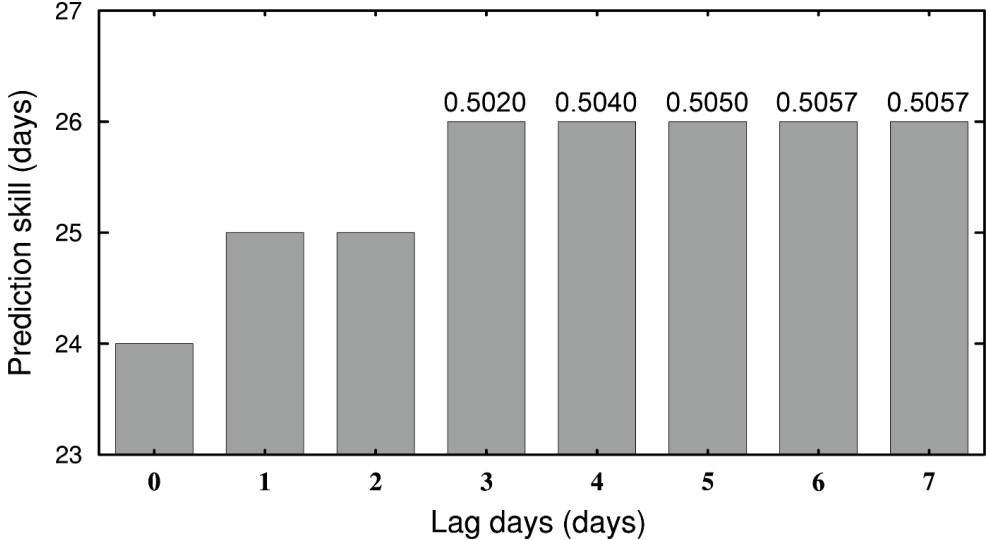

**Figure A3. MJO forecast skill of the ensemble mean of time-lagged members as a function of lag days. The values on the bars**
**represent the ACC on day 26.**





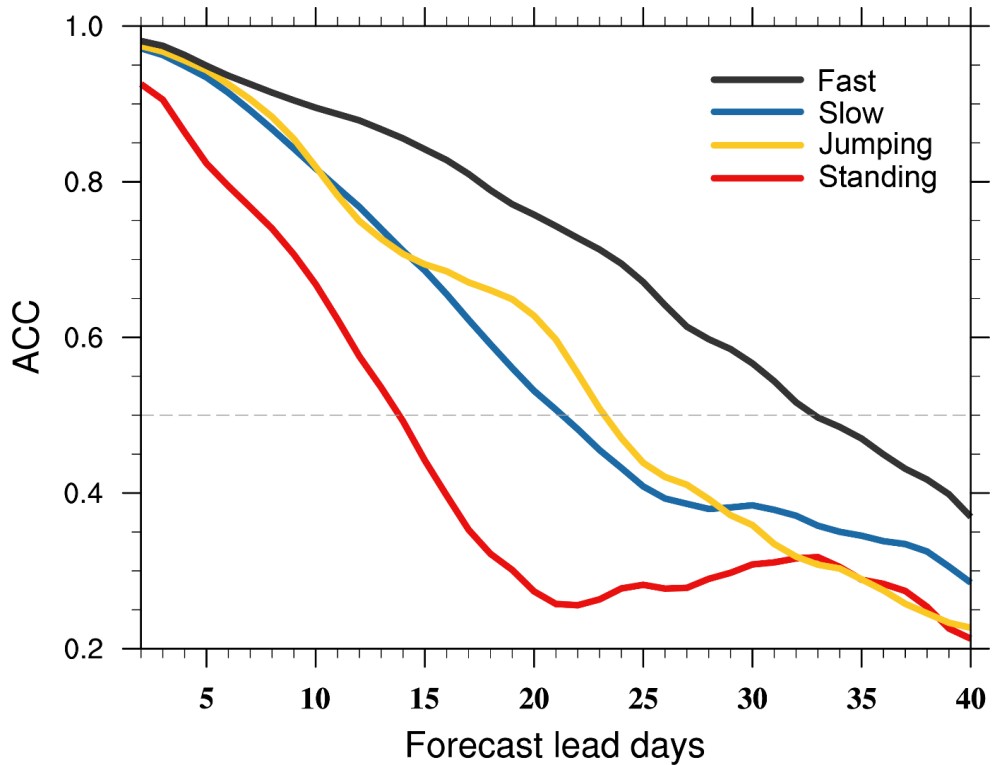


**Figure A4. The bivariate ACC as a function of forecast lead days for fast, slow, jumping, and standing MJO events. The dashed line**

**has the value of 0.5.**

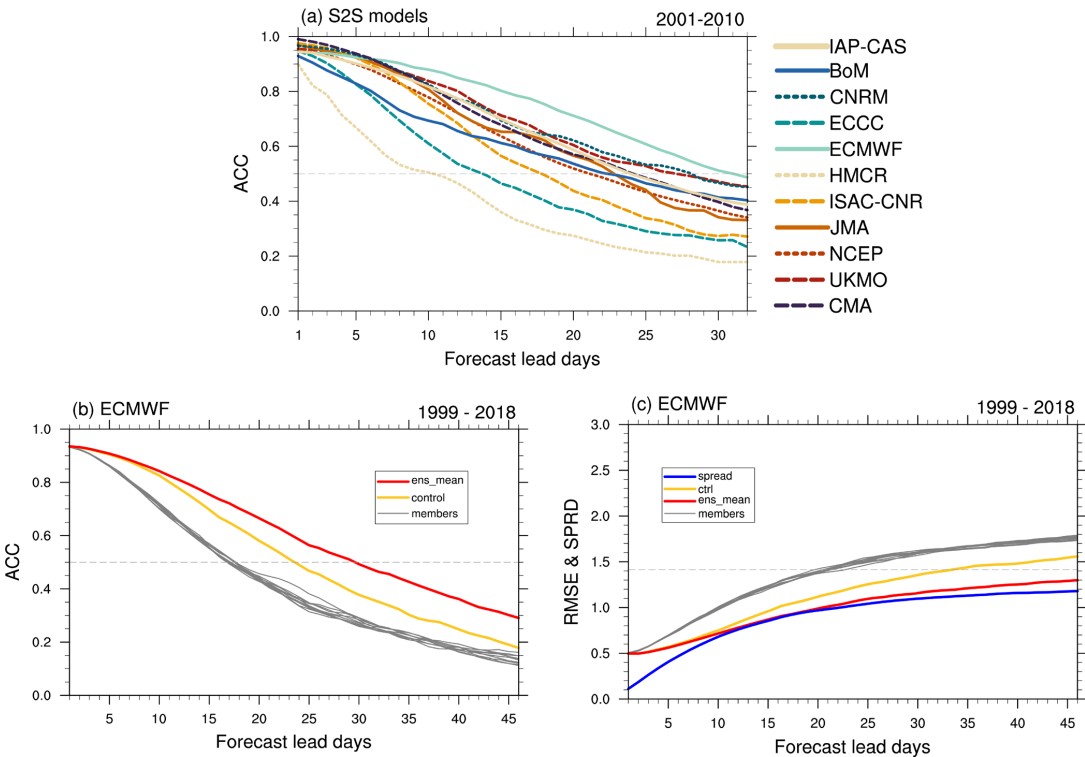

**Figure A5. MJO forecast skill of S2S models. (a) The ACC between the model ensemble means and ERA-Interim over 10-year (2001-2010) RMM index for 11 S2S models. The datasets used in this study are from the following versions (with the year of update as the version number): IAP-2022, BoM-2014, CNRM-2019, ECCC-2022, ECMWF-2022, HMCR-2022, ISAC-2017, JMA-2020, NCEP-2011, UKMO-2022, CMA-2022. (b) The ACC and (c) the RMSE from individual members (gray solid line), ensemble ctrl (green solid line), and 10-member ensemble mean (red solid line) as a function of forecast lead days. The blue solid line denotes the ensemble spread. The dashed line in (a) and (b) has the values of 0.5, and it represents 1.414 in (c).**



**Code availability**

The code of the IAP-CAS model is archived on Zenodo (https://doi.org/10.5281/zenodo.10791355). The code used to reproduce the figures in this work can be obtained from https://doi.org/10.5281/zenodo.10817813.

**Data availability**

The boundary conditions and input data are available at https://doi.org/ 10.5281/zenodo.10820243. The data for initialization in the IAP-CAS S2S system is available at http://rda.ucar.edu/datasets/ds083.2, ds083.2|DOI: 10.5065/D6M043-C6 (FNL), https://www.ncei.noaa.gov/products/optimum-interpolation-sst (NOAA OISST) and https://www.ncei.noaa.gov/products/weather-climate-models/global-forecast (GFS weather forecast). The hindcast dataset of the IAP-CAS S2S system used in the article is publicly available on the three S2S Data Portals (ECMWF: https://apps.ecmwf.int/data-sets/; CMA: http://s2s.cma.cn/index; IRI: https://iridl.ldeo.columb-ia.edu/SOURCES/ECMWF/S2S/). All the validation data are available to download from the cited references or data links shown in Section 3.1.

**Author contribution**

Q.B. led the IAP-CAS model development. All other co-authors contributed to it. B.H. and X.F.W. designed the experiments and carried them out. Y.K.L. utilized the dataset to assess the performance of the IAP-CAS S2S system and wrote the final document with contributions from all other authors. Q.B. reviewed and edited the manuscript. G.X.W., Y.M.L., and J.Y. supervised and supported this research and gave important opinions.

**Competing interests**

The authors declare no conflict of interest.

**Acknowledgments**

This work was supported by funding from the National Natural Science Foundation of China (Grant 42175161, 42261144671), and the Alliance of International Science Organizations (Grant ANSO-CR-KP-2020-01)

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
