# Peer review of "Dynamical MJO forecasts using an ensemble subseasonal-to-seasonal"

_EGUsphere, 2024_

## Referee Comment (RC2)

Review of "Dynamic MJO forecasts using an ensemble subseasonal-to-seasonal forecast system of IAP-CAS model"

For publication in *Geoscientific Model Development*

Recommendation: Major revisions

This manuscript describes a new subseasonal ensemble prediction system at IAP-CAS. The development and analysis are sound. The system itself isn't distinguished from existing S2S systems, although the error analysis is interesting and there is a nice simple method to initialize the model. In particular the analysis of convective heating in Figure 9 is a unique and very nice contribution. I have a few questions about the analyses that I would like the authors to answer before the paper is published, and so I am giving a recommendation of "major revisions".

Breaking down the MJO into different types is a useful way to determine model biases. One thing I note from Figure 5 is that the simulated "jumping" MJOs look much more like the simulated slow MJOs than the observed jumping structure. Is part of the difficulty simulating these modes that the model lacks the ability to correctly simulate the distinct jumping and slow modes? Related, the composite simulated MJO structures shown in Figure 6 appear to be significantly disrupted; most notably, the fast mode has a strange equatorial dry anomaly and enhanced westerlies at day 20 extending all the way to nearly 180 longitude. This appears to be something like an anomalous westerly wind burst which could in turn have impacts on ENSO prediction. Could the authors comment on what this physically might represent, or is it a model anomaly that is challenging to explain?

The authors' attribution of the MJO propagation errors and over-convection to a moist bias is certainly plausible and supported by the evidence. What I find strange is that the moisture bias is smallest near the Maritime Continent but the MC barrier is still a challenge for these simulations. Is the MC barrier a separate problem for the model not directly related to the propagation biases? Also the authors believe that a positive bias in evaporative fluxes could be causing the positive moisture bias; is it possible to check this?

The target analysis in Figure 4b is a nice alternate way of assessing the predictive skill of the system. Can this be interpreted as a prediction of the *development* of the MJO and/or even of the genesis of MJO events?

Minor comments

- There is too much detail in the tables. I think tables 2–5 are unnecessary for the body of the paper and can be moved into the supplemental data.
- Lines 49–51: I didn't understand the statement "models that exhibited lower forecast skills in [CMIP5] have demonstrated noteworthy improvements in the simulation of MJO". Did you mean that models with poor MJOs in CMIP5 had significantly improved MJOs in

CMIP6? Note that these should not be the skill of MJO forecasts since CMIP simulations are long uninitialized simulations.
- Section 2.1: FV3 only needs the semi-implicit sound wave solver if run nonhydrostatic; I believe these simulations are hydrostatic as are most FV3-based climate models.
- Section 2.2: That the S2S forecasts are nudged to GFS forecasts for the first ten days is interesting. Is this to avoid coupling shock at initialization?
- Section 2.4: Am I correct that there are 4 ensemble members initialized each day over a 20-year period? This would then be a very large dataset.
- Section 3.3: What is "silhouette clustering"?
- Equation 11: Is this calculation used to compute condensational heating in both the model and observations? Why was the condensational heating output by ERA5 and the model not used? (I understand if this was not output from the model, and ERA's condensational heating estimate may be skewed by the data assimilation used by the reanalysis.)
- Figure 14: Is this the same as Figure 6, but the shading is Q850 instead of precipitation?
- Figure A2: I see FGOALS-f2 is the model shown in this paper. Are the other models shown here earlier versions of FGOALS, and in what order were they developed?

For the most part, the article is well-written. I do see some English usage that could be improved. Here are some examples:
- Title "Dynamic" → "Dynamical" and "of IAP-CAS" → "of the IAP-CAS"
- Line 66: "low" → "lower"
- Lines 78–82: recommend using future or present tense instead of past tese in this paragraph.
- Line 234: "inconsecutive movement"; do you mean discontinuous propagation/movement?
- Line 235: "coupling" → "coupled.

Lucas Harris

---

## Author Comment (AC1)

Dear Prof. Liu,

We greatly appreciate your insightful and helpful comments regarding our manuscript, especially concerning the physical explanation for the rapid propagation speed of MJO in the model. We have carefully revised the manuscript in response to your comments, primarily focusing on Sections 5 and 6, as well as addressing various specific details. Below are the point-by-point replies to your comments and concerns.

**Comment 1:**

*It appears that the IAP-CAS model has already been involved in the S2S project, and we can download the hindcast data from the S2S project. Is the model used in this work the same version as the one from the S2S project?*

**Response:**

Yes, the hindcast data available on the S2S website and the model described in the manuscript are indeed the same version (IAP-CAS-V1.3). Our team's research has found that incorporating moisture initialization has improved the model's skill, as referenced in the following article. Therefore, real-time forecast data has now transitioned to the new version (with moisture initialization, increasing ensemble size to 16 members, IAP-CAS-V1.4). We're going to send the hindcast GRIB datasets of the new version to the S2S data archive.

Zeng, L., Bao, Q., Wu, X., He, B., Yang, J., Wang, T., Liu, Y., Wu, G., and Liu, Y.: Impacts of humidity initialization on MJO prediction: A study in an operational sub-seasonal to seasonal system, Atmospheric Research, 294, 106946, https://doi.org/10.1016/j.atmosres.2023.106946, 2023.

**Comment 2:**

*Why not create a figure comparing the prediction skill among all S2S models? Some studies have conducted such comparisons, and it is necessary to present the average and best skill among the current S2S models.*

**Response:**

Your suggestion is excellent, and we have compared the prediction skills of 12 S2S models for MJO in the Appendix (Figure A4), using the latest versions of each model. We have revised the figure to include comparisons across multiple models and their different versions. Additionally, we have also assessed the skill of deterministic forecasts versus ensemble forecasts. The modified figure is provided below.

[Figure]

The MJO index of 12 S2S models and ERA-Interim from the S2S website (http://www.s2sprediction.net/) is used for evaluation during the standard hindcast period 2001-2010, except for CMA, which spans from 2008 to 2013. The solid lines represent the skill of ensemble mean forecasts, while the dashed lines represent the skill of deterministic forecasts.

**Comment 3:**

*Was the prediction skill of 24 days calculated for the annual MJO or for the boreal winter MJO? It is important to clearly state whether the main conclusions are for the annual mean or for the boreal winter. (Sometimes you show the results for the annual mean, while some figures were drawn for the boreal winter), as S2S models exhibit a significant annual cycle in the prediction of MJO.*

**Response:**

The prediction skill of 24 days is calculated for the annual MJO. In section 4 of the paper, our evaluation primarily focuses on the annual MJO, whereas sections 5 and 6 delve further into different types of boreal winter MJO events. We have included clarifications in the revised manuscript, which can be found in lines 232 and 268, as well as in the captions of Figures 3 and 5.

**Comment 4:**

*Lines 168-170, you only have 16 ensembles since 2019, while in Fig. 3, you also presented 16 ensembles for the long period of 1999-2018.*

**Response:**

The mention of "Since June 1st, 2019, the IAP-CAS S2S system has been operating 16 ensemble members daily' in lines 168-170 refers to real-time forecast results, whereas only four ensembles are available for hindcast data. However, in Figure 3(c), we employed the time-lagged method to increase the ensemble size from 4 to 16, primarily to assess the extent to which the time-lag method enhances MJO skill. Appropriate clarification has been added in the paper to prevent misunderstandings, as indicated in the caption of Figure 3.

**Comment 6:**

*Line 42: The impact of MJO on sub-subseasonal prediction of each sub-monsoon precipitation has been well discussed (Liu et al., 2022), and should be referenced.*

**Response:**

We appreciate the suggestion to reference the work of Liu et al. (2022), which discusses the impact of MJO on the sub-subseasonal prediction of each sub-monsoon precipitation. We have incorporated this reference into our manuscript accordingly, as seen in line 43 of the revised manuscript.

**Comment 7:**

*Lines 117-119: There were many phenomena that affect the MJO propagation. I suggest deleting this statement as it is not directly related to this work.*

**Response:**

Thank you for your suggestion. The statement has been removed.

**Comment 8:**

*Lines 145: I cannot follow why you use the 10-day forecast nudging from the GFS forecast. Should we attribute the good prediction skill of 24 days to IAP-CAS or GFS?*

**Response:**

Thank you for your feedback. Our aim is to improve S2S forecast skills, which require different strategies than traditional climate predictions. According to general consensus, numerical weather predictions are more accurate than climate predictions for the first 10 days. While we use 10-day GFS weather forecast data to enhance S2S forecasts, our experiments have revealed that the optimal nudging duration is 5 days (Zeng et al. 2023). This adjustment will be implemented in our next-generation S2S system.

**Comment 9:**

*Lines 245: Was this underdispersive due to weak initial perturbation of the time-lag method?*

**Response:**

Yes, we think so. Hence, we are also considering adopting a new ensemble generation method to address this issue.

**Comment 10:**

*Lines 306-310: You can calculate the phase speed in this Hovmöller diagram directly.*

**Response:**

Thank you for your suggestion! We now directly calculate the phase speed in this Hovmöller diagram, as shown in the figure below. We have also made corresponding modifications in Section 5 to align with the adjustments.

[Figure]

**Comment 11:**

*I have a different explanation for the phase speed difference. In Fig. 6, the predicted zonal scale of the MJO, represented by the easterly wind anomalies to the east of the MJO convective center, covers a larger region than observed, which is more obvious for the slow-propagating mode. The moist central Pacific in IAP-CAS overestimates the zonal scale of the MJO, which will increase the eastward propagation speed of the MJO, since the phase speed is inversely proportional to the wave number, as shown in previous work (Wang et al. 2019Sci. Adv. Diversity of MJO). The increased MSE tendency to the east of the MJO can explain the increased amplitude of the MJO, rather than the propagation speed. Let's make an assumption: for the same speed, the stronger MJO also has a larger MSE tendency to the east than the weaker MJO.*

**Response:**

Thank you for your insightful perspective. After careful consideration, we agree with your explanation regarding the phase speed difference. We appreciate your suggestion to supplement our analysis in Section 6 accordingly. In this section, we have elaborated on how moisture theory can elucidate the development of intensity in propagating MJO events, thereby leading to an amplification in the intensity and zonal scale of the coupled wave, consequently resulting in the observed faster MJO propagation speed.

We sincerely appreciate the time and effort invested by the reviewers in evaluating our manuscript. We look forward to any additional feedback or suggestions.

Thank you and best regards.

Sincerely,

Qing Bao

State Key Laboratory of Numerical Modeling for Atmospheric Sciences and Geophysical Fluid Dynamics (LASG), Institute of Atmospheric Physics, Chinese Academy of Sciences, Beijing 100029, China

---

## Author Comment (AC2)

Dear Dr. Lucas Harris,

Thank you very much for your detailed comments and suggestions on this manuscript. Your feedback has been tremendously helpful to us. Below are our responses and the modifications made in light of some of the issues you raised.

**Broad comments:**

**Comment 1:**
*Breaking down the MJO into different types is a useful way to determine model biases. One thing I note from Figure 5 is that the simulated "jumping" MJOs look much more like the simulated slow MJOs than the observed jumping structure. Is part of the difficulty simulating these modes that the model lacks the ability to correctly simulate the distinct jumping and slow modes?*

**Response:**
Thank you for your constructive comments about the jumping and slow modes of the MJO. Since Figure 5 depicts zonal averages, some nuances may indeed be obscured. However, upon closer examination of the Evolution patterns for the jumping MJO (see the figure below), it becomes evident that the model simulations also capture signals indicative of "jumping". By the fifth day, we observe interruptions in convective signals over the MC region. However, due to potentially stronger convective activity, these signals may not be as pronounced in the zonal averages depicted in the Hovmöller diagram. Thus, while the distinction may not be as clear in Figure 5, the model does exhibit characteristics of the jumping MJO, albeit potentially overshadowed in zonal averages. We have revised the manuscript (see lines 317-319 and Figure A3 of the revised manuscript) and clarified these points further.

[Figure]

**Comment 2:**

*The composite simulated MJO structures shown in Figure 6 appear to be significantly disrupted; most notably, the fast mode has a strange equatorial dry anomaly and enhanced westerlies at day 20 extending all the way to nearly 180° longitude. This appears to be something like an anomalous westerly wind burst which could in turn have impacts on ENSO prediction. Could the authors comment on what this physically might represent, or is it a model anomaly that is challenging to explain?*

**Response:**

We think this phenomenon is also related to the model's bias in moisture. As mentioned in the revised manuscript (Lines 324-333), a positive moisture bias may lead to enhanced convection. During the convective development phase, this results in strengthened easterly winds dominating the convective activity, thus accelerating the MJO's propagation. Conversely, after the MJO traverses the MC region, during its decaying phase, the persistently strong convection also leads to intensified westerly winds dominating the convective activity in the western part, hastening the MJO's decay process (Page 14 of the revised manuscript).

**Comment 3:**

*The authors' attribution of the MJO propagation errors and over-convection to a moist bias is certainly plausible and supported by the evidence. What I find strange is that the moisture bias is smallest near the Maritime Continent but the MC barrier is still a challenge for these simulations. Is the MC barrier a separate problem for the model not directly related to the propagation biases?*

**Response:**

In the IAP-CAS model, although some moisture bias is present, we can relatively accurately capture the gradient of background moisture over the Maritime Continent during the winter (see Figure 10). As a result, the IAP-CAS model performs relatively well in simulating the propagation of the MJO over the Maritime Continent region. However, it may lead to MJO decay due to the robust development of the simulated propagating MJO negative phase after crossing the MC region (see lines 305-316 of the revised manuscript). Many other models may still encounter challenges with the "MC barrier" due to inaccuracies in simulating the gradient of background moisture over the Maritime Continent during the winter, as highlighted by Gonzalez and Jiang (2017).

[Figure]

**Figure 10. The winter (November–April) mean specific humidity (g kg$^{-1}$) on 850hPa for (a) observation and (b) IAP-CAS model.**

[Figure]

a) ERA-Interim

b) Good MJO Models

c) Poor MJO Models

g kg$^{-1}$

2.5  3.5  4.5  5.5  6.5  7.5  8.5  9.5  10.5

Gonzalez, A. O. and Jiang, X.: Winter mean lower tropospheric moisture over the Maritime Continent as a climate model diagnostic metric for the propagation of the Madden-Julian oscillation, Geophysical Research Letters, 44, 2588–2596, https://doi.org/10.1002/2016GL072430, 2017.

**Comment 4:**

*The authors believe that a positive bias in evaporative fluxes could be causing the positive moisture bias; is it possible to check this?*

**Response:**

After reviewing your suggestions, we investigated the potential cause in IAP-CAS model control run and summarized our findings in the figure below. As depicted in the figure below, during the boreal winter (December, January, and February, in short of DJF), when comparing with ERA40 data, we utilize the Surface Latent Heat Flux variable, which to some extent reflects the evaporation conditions. In the equatorial regions, it is evident that the evaporation in the IAP-CAS model is significantly larger.

[Figure]

**Surface latent heat flux in the IAP-CAS model**

**Comment 5:**

*The target analysis in Figure 4b is a nice alternate way of assessing the predictive skill of the system. Can this be interpreted as a prediction of the development of the MJO and/or even of the genesis of MJO events?*

**Response:**

The target analysis involves selecting cases where the amplitude of the MJO exceeds 1 and analyzing the forecasts for the 65 days preceding these cases. This analysis serves as a metric for evaluating the forecasting system's performance in capturing MJO event

precursors. When combined with the initial analysis, we gain insight into the forecast skill before and after the occurrence of strong and weak MJO events.

**# Minor comments:**

**Comment 1:**

*There is too much detail in the tables. I think tables 2–5 are unnecessary for the body of the paper and can be moved into the supplemental data.*

**Response:**

Thank you for your thoughtful suggestion. We have now revised the manuscript accordingly, moving tables 2–5 into the supplemental data as per your recommendation. Your feedback is greatly appreciated.

**Comment 2:**

*Lines 49–51: I didn't understand the statement "models that exhibited lower forecast skills in [CMIP5] have demonstrated noteworthy improvements in the simulation of MJO". Did you mean that models with poor MJOs in CMIP5 had significantly improved MJOs in CMIP6? Note that these should not be the skill of MJO forecasts since CMIP simulations are long uninitialized simulations.*

**Response:**

From CMIP5 to CMIP6, certain models exhibited a notable enhancement in their MJO skills, as well as the IAP-CAS model. The IAP-CAS model in CMIP5 is FGOALS-s2, while in CMIP6 is FGOALS-f3-L/H. (see the figure below). Here, the emphasis lies on improvements in the physical frameworks of the models for simulating MJO, excluding the impact of initialization.

[Figure]

Chen, G., Ling, J., Zhang, R., Xiao, Z., and Li, C.: The MJO From CMIP5 to CMIP6: Perspectives From Tracking MJO Precipitation, Geophysical Research Letters, 49, https://doi.org/10.1029/2021GL095241, 2022.

**Comment 3:**

*Section 2.1: FV3 only needs the semi-implicit sound wave solver if run nonhydrostatic; I believe these simulations are hydrostatic as are most FV3-based climate models.*

**Response:**

Yes, we fully agree with you. The FV3 dynamic core in the IAP-CAS model is currently used by the hydrostatic solver. This choice is based on the IAP-CAS model's resolution of C96, where the difference between hydrostatic and nonhydrostatic solvers can be neglected. However, a nonhydrostatic solver entails higher computational costs. Therefore, we selected the hydrostatic solver. Nevertheless, as we plan to enhance the model resolution from C96 to C384, we'll certainly switch to the nonhydrostatic solver.

**Comment 4:**

*Section 2.2: That the S2S forecasts are nudged to GFS forecasts for the first ten days is interesting. Is this to avoid coupling shock at initialization?*

**Response:**

Indeed, that is one of the reasons. Additionally, we know that the S2S forecast lies between weather prediction and climate projection, influenced by both initial conditions and external forcing. Initially, S2S prediction skills lag behind those of weather forecasts for a certain period. Thus, we aim to enhance the quality of initial forecasts in S2S by nudging GFS weather forecast data to ultimately improve S2S prediction accuracy. In our system (the IAP-CAS model v1.3) described in this manuscript, we employed 10 days of GFS data for nudging. However, through further investigation, we have found that perhaps the optimal nudging duration is 5 days (Zeng et al., 2023). This refinement will be integrated into our forthcoming S2S system upgrade (the IAP-CAS model v1.4).

we have updated the manuscript with these explanations. Please refer to lines 148-151 in the revised manuscript.

Zeng, L., Bao, Q., Wu, X., He, B., Yang, J., Wang, T., Liu, Y., Wu, G., and Liu, Y.: Impacts of humidity initialization on MJO prediction: A study in an operational sub-seasonal to seasonal system, Atmospheric Research, 294, 106946, https://doi.org/10.1016/j.atmosres.2023.106946, 2023.

**Comment 5:**

*Section 2.4: Am I correct that there are 4 ensemble members initialized each day over a 20-year period? This would then be a very large dataset.*

**Response:**

You're right, this is indeed a rather large dataset, with a total size of approximately 11TB. Evaluating this dataset, I believe, would greatly benefit the further enhancement of the model. The details regarding the dataset size have also been updated in the revised manuscript. Please refer to lines 116-119.

**Comment 6:**

*Section 3.3: What is "silhouette clustering"?*

**Response:**

"Silhouette clustering" is a technique used in cluster analysis to assess the quality of clustering results. It calculates a silhouette coefficient for each data point, which measures how similar that point is to its own cluster compared to other clusters.

Essentially, it helps to determine the appropriateness of the clustering by quantifying the compactness and separation of clusters. Details are in the following paper and we have also updated the necessary references in the revised manuscript, see lines 255-256.

**Rousseeuw, P.: Silhouettes - a Graphical Aid to the Interpretation and Validation of Cluster-Analysis, J. Comput. Appl. Math., 20, 53–65, https://doi.org/10.1016/0377-0427(87)90125-7, 1987.**

**Comment 7:**

*Equation 11: Is this calculation used to compute condensational heating in both the model and observations? Why was the condensational heating output by ERA5 and the model not used? (I understand if this was not output from the model, and ERA's condensational heating estimate may be skewed by the data assimilation used by the reanalysis.)*

**Response:**

Yes, condensational heating is indeed computed both in the model and observations. The purpose of this approach is to mitigate the influence of other factors, including the potential impact of data assimilation, and ensure an equitable comparison between the model and observations.

**Comment 8:**

*Figure 14: Is this the same as Figure 6, but the shading is Q850 instead of precipitation?*

**Response:**

Yes, in this figure, we aimed to illustrate that besides the winter mean moisture state, MJO-related moisture anomalies also have a significant impact on MJO prediction. Adding wind vectors allows us to more intuitively observe the advection of MJO-related moisture anomalies, even though this wind field is the same as in Figure 6 (which has been modified and referenced as Figure 7 in the revised manuscript).

**Comment 9:**

*Figure A2: I see FGOALS-f2 is the model shown in this paper. Are the other models shown here earlier versions of FGOALS, and in what order were they developed?*

**Response:**

These models don't correspond to a simple new-to-old version relationship. All these models originate from the Institute of Atmospheric Physics, Chinese Academy of Sciences. FGOALS-g1, FGOALS-g2, and FGOALS-g3 represent successive versions of the Flexible Global Ocean-Atmosphere-Land System model Grid-point,

participating in CMIP3, CMIP5, and CMIP6, respectively. Conversely, FGOALS-f2 and FGOALS-f3 are Finite Volume versions. While FGOALS-f2 primarily focuses on seamless prediction, FGOALS-f3 emphasizes climate simulation. FGOALS-f3 also participated in CMIP6 experiments.

**Comment 10:**

*For the most part, the article is well-written. I do see some English usage that could be improved. Here are some examples:*

*- Title "Dynamic" → "Dynamical" and "of IAP-CAS" → "of the IAP-CAS"*

*- Line 66: "low" → "lower"*

*- Lines 78–82: recommend using future or present tense instead of past tense in this paragraph.*

*- Line 234: "inconsecutive movement"; do you mean discontinuous propagation/movement?*

*- Line 235: "coupling" → "coupled.*

**Response:**

Thanks for your careful checks. We are sorry for our carelessness. Based on your comments, we made the corrections to make the word harmonized within the whole manuscript.

We sincerely appreciate the time and effort invested by the reviewers in evaluating our manuscript. We look forward to any additional feedback or suggestions.

Thank you and best regards.
Sincerely,

Qing Bao
State Key Laboratory of Numerical Modeling for Atmospheric Sciences and Geophysical Fluid Dynamics (LASG), Institute of Atmospheric Physics, Chinese Academy of Sciences, Beijing 100029, China

---

## Author Response (AR2)

Dear Editors and Referees,

Thanks for your detailed review and constructive feedback provided on our manuscript. Your insightful comments have been invaluable in enhancing the quality of our work. In response to Dr. Lucas Harris's concerns regarding the hydrostatic solver of FV3, we have made the necessary revisions. The following are detailed comment and reply.

**Comment:**

*The manuscript has been notably improved compared to the original version, and my comments have been answered to my satisfaction. One clarification that is still needed in the manuscript is that the IAP-CAS model is using the hydrostatic solver, which is indeed the best choice at this resolution. In the hydrostatic solver there are no vertically-propagating sound waves and so the semi-implicit solver is not needed. I would replace the sentence on lines 107 & 108 with a statement that the hydrostatic solver of FV3 is used.*

**Response:**

Thank you for helping us identify this issue. We have clarified that our model uses the hydrostatic solver of FV3 to replace the inaccurate information. The specific changes can be found on lines 107-109 of the revised paper.

Thank you for your guidance and support again!

Best regards,

Qing Bao
State Key Laboratory of Numerical Modeling for Atmospheric Sciences and Geophysical Fluid Dynamics (LASG), Institute of Atmospheric Physics, Chinese Academy of Sciences, Beijing 100029, China